# E proteins sharpen neurogenesis by modulating proneural bHLH transcription factors' activity in an E-box-dependent manner

Gwenvael Le Dréau[1†]*, René Escalona[1†‡], Raquel Fueyo[2], Antonio Herrera[1], Juan D Martínez[1], Susana Usieto[1], Anghara Menendez[3], Sebastian Pons[3], Marian A Martinez-Balbas[2], Elisa Marti[1]

[1]Department of Developmental Biology, Instituto de Biología Molecular de Barcelona, Barcelona, Spain; [2]Department of Molecular Genomics, Instituto de Biología Molecular de Barcelona, Barcelona, Spain; [3]Department of Cell Biology, Instituto de Biología Molecular de Barcelona, Barcelona, Spain

*For correspondence:
gldbmc@ibmb.csic.es

[†]These authors contributed equally to this work

Present address: [‡]Departamento de Embriología, Facultad de Medicina, Universidad Nacional Autónoma de México, México City, Mexico

Competing interests: The authors declare that no competing interests exist.

**Abstract** Class II HLH proteins heterodimerize with class I HLH/E proteins to regulate transcription. Here, we show that E proteins sharpen neurogenesis by adjusting the neurogenic strength of the distinct proneural proteins. We find that inhibiting BMP signaling or its target ID2 in the chick embryo spinal cord, impairs the neuronal production from progenitors expressing ATOH1/ASCL1, but less severely that from progenitors expressing NEUROG1/2/PTF1a. We show this context-dependent response to result from the differential modulation of proneural proteins' activity by E proteins. E proteins synergize with proneural proteins when acting on CAGSTG motifs, thereby facilitating the activity of ASCL1/ATOH1 which preferentially bind to such motifs. Conversely, E proteins restrict the neurogenic strength of NEUROG1/2 by directly inhibiting their preferential binding to CADATG motifs. Since we find this mechanism to be conserved in corticogenesis, we propose this differential co-operation of E proteins with proneural proteins as a novel though general feature of their mechanism of action.
DOI: https://doi.org/10.7554/eLife.37267.001

## Introduction

The correct functioning of the vertebrate central nervous system (CNS) relies on the activity of a large variety of neurons that can be distinguished by their morphologies, physiological characteristics and anatomical locations (*Zeng and Sanes, 2017*). Such heterogeneity is generated during the phase of neurogenesis, once neural progenitors have been regionally specified and are instructed to exit the cell cycle and differentiate into discrete neuronal subtypes (*Guillemot, 2007*).

Neuronal differentiation and subtype specification are brought together by a small group of transcription factors (TFs) encoded by homologues of the *Drosophila* gene families *Atonal, Achaete-Scute, Neurogenins/dTap* and *p48/Ptf1a/Fer2* (*Bertrand et al., 2002*; *Huang et al., 2014*). These TFs represent a subgroup of the class II of helix-loop-helix proteins and all share a typical basic helix-loop-helix (bHLH) structural motif, where the basic domain mediates direct DNA binding to CANNTG sequences (known as E-boxes) and the HLH region is responsible for dimerization and protein-protein interactions (*Massari and Murre, 2000*; *Bertrand et al., 2002*). They are generally expressed in mutually exclusive populations of neural progenitors along the rostral-caudal and dorsal-ventral axes (*Gowan et al., 2001*; *Lai et al., 2016*). They are typically referred to as proneural proteins, since they are both necessary and sufficient to switch on the genetic programs that drive

**eLife digest** The brain and spinal cord are made up of a range of cell types that carry out different roles within the central nervous system. Each type of neuron is uniquely specialized to do its job. Neurons are produced early during development, when they differentiate from a group of cells called neural progenitor cells. Within these groups, molecules called proneural proteins control which types of neurons will develop from the neural progenitor cells, and how many of them.

Proneural proteins work by binding to specific patterns in the DNA, called E-boxes, with the help of E proteins. E proteins are typically understood to be passive partners, working with each different proneural protein indiscriminately. However, Le Dréau, Escalona et al. discovered that E proteins in fact have a much more active role to play.

Using chick embryos, it was found that E proteins influence the way different proneural proteins bind to DNA. The E proteins have preferences for certain E-boxes in the DNA, just like proneural proteins do. The E proteins enhanced the activity of the proneural proteins that share their E-box preference, and reined in the activity of proneural proteins that prefer other E-boxes. As a result, the E proteins controlled the ability of these proteins to instruct neural progenitor cells to produce specific, specialized neurons, and thus ensured that the distinct types of neurons were produced in appropriate amounts.

These findings will help shed light on the roles E proteins play in the development of the central nervous system, and the processes that control growth and lead to cell diversity. The results may also have applications in the field of regenerative medicine, as proneural proteins play an important role in cell reprogramming.

DOI: https://doi.org/10.7554/eLife.37267.002

pan-neuronal differentiation and neuronal subtype specification during development (*Guillemot, 2007*). This unique characteristic is also illustrated by their ability to reprogram distinct neural and non-neural cell types into functional neurons (*Masserdotti et al., 2016*).

Regulating the activity of these proneural proteins is crucial to ensure the production of appropriate numbers of neurons without prematurely depleting the pools of neural progenitors. In cycling neural progenitors, the transcriptional repressors HES1 and HES5 act in response to Notch signalling to maintain proneural TF transcripts oscillating at low levels (*Imayoshi and Kageyama, 2014*). The proneural proteins are also regulated at the post-translational level. Ubiquitination and phosphorylation have been reported to control their stability, modify their DNA binding capacity or even terminate their transcriptional activity (*Ali et al., 2011*; *Li et al., 2012*; *Ali et al., 2014*; *Quan et al., 2016*). Furthermore, the activity of these proneural proteins is highly dependent on protein–protein interactions, and particularly on their dimerization status. It is generally admitted that these TFs must form heterodimers with the more broadly expressed class I HLH/E proteins to produce their transcriptional activity (*Wang and Baker, 2015*). In this way, the activity of proneural proteins can be controlled by upstream signals that regulate the relative availability of E proteins. Members of the Inhibitor of DNA binding (ID) family represent such regulators. As they lack the basic domain required for direct DNA-binding, ID proteins sequester E proteins through a physical interaction and thereby produce a dominant-negative effect on proneural proteins (*Massari and Murre, 2000*; *Wang and Baker, 2015*). Hence, several sophisticated regulatory mechanisms are available during development to control proneural protein activity and fine-tune neurogenesis.

Bone morphogenetic proteins (BMPs) contribute to multiple processes during the formation of the vertebrate CNS (*Liu and Niswander, 2005*; *Le Dréau and Martí, 2013*). Yet it is only in the past few years that their specific role in controlling vertebrate neurogenesis has begun to be defined (*Le Dréau et al., 2012*; *Segklia et al., 2012*; *Choe et al., 2013*; *Le Dréau et al., 2014*). During spinal cord development, SMAD1 and SMAD5, two canonical TFs of the BMP pathway (*Massagué et al., 2005*), dictate the mode of division that spinal progenitors adopt during primary neurogenesis. Accordingly, strong SMAD1/5 activity promotes progenitor maintenance while weaker activity enables neurogenic divisions to occur (*Le Dréau et al., 2014*). This model explains how inhibition of BMP7 or SMAD1/5 activity provokes premature neuronal differentiation and the concomitant depletion of progenitors. However, it does not explain why the generation of distinct subtypes

of dorsal interneurons are affected differently (*Le Dréau et al., 2012*), nor how BMP signaling affects the activity of the proneural proteins expressed in the corresponding progenitor domains.

Here, we have investigated these questions, extending our analysis to primary spinal neurogenesis along the whole dorsal-ventral axis. As such, we identified a striking correlation between the requirement of canonical BMP activity for the generation of a particular neuronal subtype and the proneural protein expressed in the corresponding progenitor domain. Inhibiting the activity of BMP7, SMAD1/5 or their downstream effector ID2 strongly impaired the production of neurons by spinal progenitors expressing either ATOH1 or ASCL1 alone, while it had a much weaker effect on the generation of the neuronal subtypes derived from progenitors expressing NEUROG1, NEUROG2 or PTF1a. We found that this differential responsiveness originates from an E-box dependent mode of co-operation of the class I HLH/E proteins with the proneural proteins. E proteins interact with proneural proteins to aid their interaction with CAGSTG E-boxes, facilitating the ability of ASCL1 and ATOH1 to promote neurogenic divisions and hence, neuronal differentiation. Conversely, E proteins inhibit proneural protein binding to CADATG motifs, consequently restraining the ability of NEUROG1/2 that preferentially bind to these motifs to trigger neurogenic division and promote neuronal differentiation. Similar results were obtained in the context of corticogenesis, suggesting that this differential co-operation of E proteins with the distinct proneural proteins is a general feature of their mode of action.

## Results

### The canonical BMP pathway differentially regulates the generation of spinal neurons derived from progenitors expressing ASCL1/ATOH1 or NEUROG1/NEUROG2/PTF1a

We previously reported that BMP7 signalling through its canonical effectors SMAD1 and SMAD5, is differentially required for the generation of the distinct subtypes of dorsal spinal interneurons (*Figure 1A* and *Le Dréau et al., 2012*). Here, we extend this analysis to the generation of neuronal subtypes produced in the ventral part of the developing chick spinal cord. Inhibiting BMP7 or SMAD1/5 expression by *in ovo* electroporation of specific sh-RNA-encoding plasmids at stage HH14-15 produced a significant reduction in the generation of p2-derived Chx10[+] (V2a) and Gata3[+](V2b) subtypes 48 hr post-electroporation (hpe), whereas Evx1[+] (V0v), En1[+] (V1) interneurons and Isl1[+] motor neurons were not significantly affected (*Figure 1—figure supplement 1*). These results revealed a correlation whereby the requirement of the canonical BMP pathway for the generation of discrete spinal neuron subtypes is linked to the proneural protein expressed in the corresponding progenitor domain (*Figure 1B,C*). The neuronal subtypes strongly affected by BMP7/SMAD1/5 inhibition (dI1, dI3, dI5: *Figure 1B,C*) were generated from spinal progenitors expressing ATOH1 (dP1) or ASCL1 alone (dP3, dP5). By contrast, all the neuronal subtypes deriving from spinal progenitors expressing either NEUROG1 alone (dP2, dP6-p1) or NEUROG2 (pMN) were much less severely affected (*Figure 1B,C*). Intriguingly, the V2a/b interneurons that display intermediate sensitivity to BMP7/SMAD1/5 inhibition are derived from p2 progenitors that express both ASCL1 and NEUROG1 (*Misra et al., 2014*), while the relatively insensitive dI4 interneurons are derived from dP4 progenitors that express PTF1a together with low levels of ASCL1 (*Figure 1B,C*, Glasgow et al. 2005).

These correlations were particularly interesting in view of recent genome-wide ChIPseq studies that identified the optimal E-box (CANNTG) motifs bound by these TFs: ATOH1 and ASCL1 both preferentially bind to CAGCTG E-boxes (*Castro et al., 2011*; *Klisch et al., 2011*; *Lai et al., 2011*; *Borromeo et al., 2014*), whereas the optimal motif for NEUROGs is CADATG (where D stands for A, G or T: see *Seo et al., 2007*; *Madelaine and Blader, 2011*). Interestingly, most of the E-boxes bound by PTF1a in the developing spinal cord correspond to the CAGCTG motif favored by ASCL1 and ATOH1, yet PTF1a can bind to the NEUROG-like CAGATG motifs in a significant proportion of its targets genes (*Borromeo et al., 2014*). These observations suggested that the sensitivity of a given progenitor domain to canonical BMP activity originates from the intrinsic DNA-binding preferences of the different proneural bHLH TFs (*Figure 1D*). In many cell contexts, BMP signaling is mediated by ID proteins (*Hollnagel et al., 1999*; *Moya et al., 2012*; *Genander et al., 2014*), which physically sequester class I HLH/E proteins to produce a dominant-negative effect on proneural

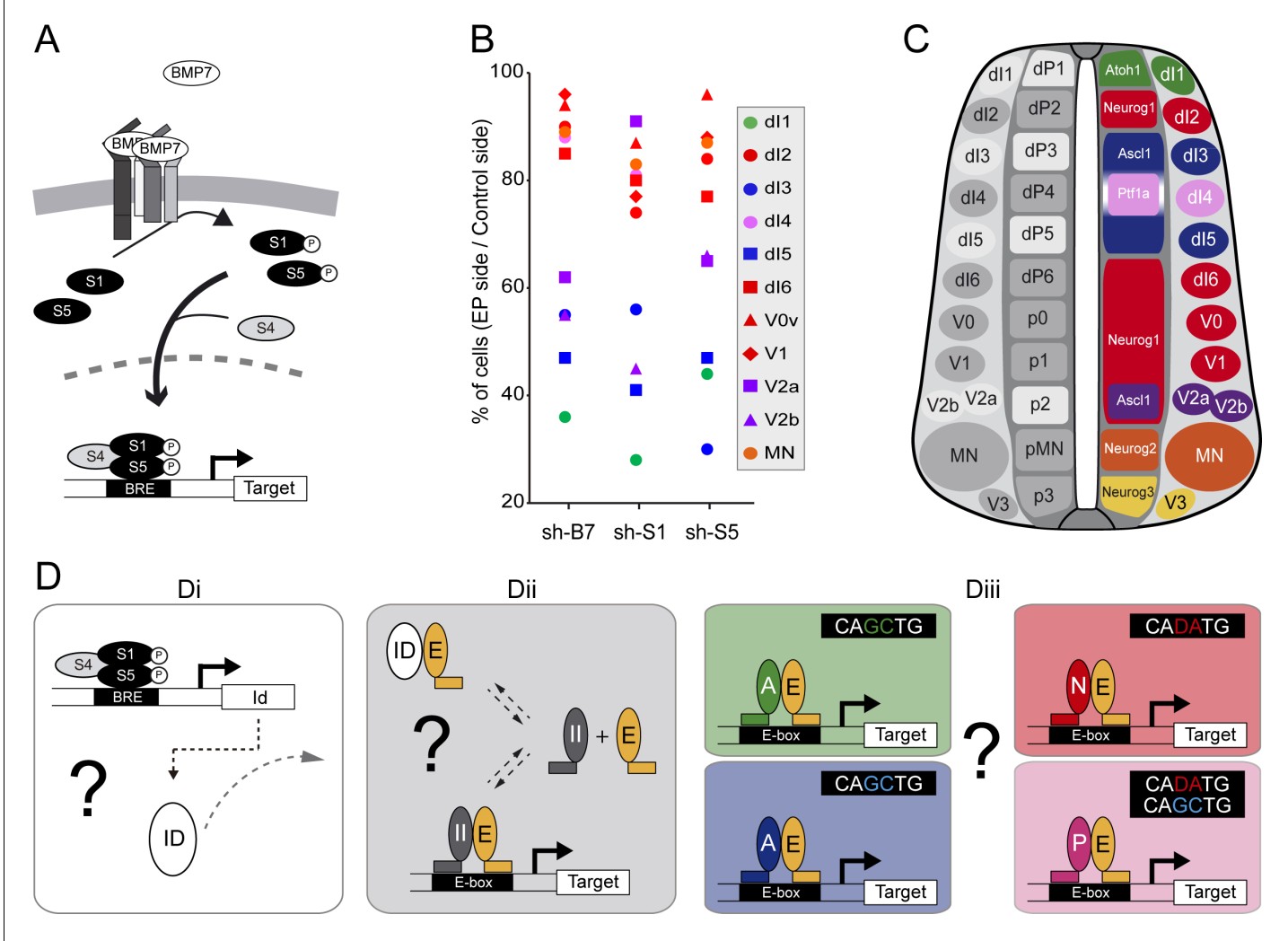

**Figure 1.** The canonical BMP pathway differentially regulates the generation of spinal neurons derived from progenitors that express ASCL1/ATOH1 or NEUROG1/NEUROG2/PTF1a. (A) Actors of the canonical BMP pathway (BMP7, SMAD1 and SMAD5) known to regulate spinal neurogenesis. (B) Dot-plot representing the spinal neuronal subtypes generated 48 hpe with plasmids producing sh-RNA targeting *cBmp7* (sh-B7), *cSmad1* (sh-S1) or *cSmad5* (sh-S5), comparing the electroporated side to the contra-lateral side. The colour code corresponds to the proneural proteins expressed in the corresponding progenitor domains, as shown in C. (C) Drawing of a transverse section of the developing spinal cord at mid-neurogenesis, highlighting: (left) the neuronal subtypes strongly (white) or moderately (grey) affected by inhibiting canonical BMP activity, and (right) a colour-coded representation of the proneural proteins expressed in the corresponding progenitor domains. (D) Working hypothesis whereby we propose to test if (i) the canonical BMP activity is mediated by ID proteins; (ii) ID proteins act by sequestering E proteins (E, orange), thereby inhibiting the activity of class II HLH/proneural proteins (II, grey); and (iii) E proteins co-operate equally or differentially with the distinct proneural proteins as a function of their preferential binding to specific E-box sequences. .

DOI: https://doi.org/10.7554/eLife.37267.003

The following figure supplement is available for figure 1:

**Figure supplement 1.** Inhibiting the canonical BMP pathway affects the generation of ventral spinal neurons.

DOI: https://doi.org/10.7554/eLife.37267.004

proteins (*Figure 1D*). While this hypothetical signaling cascade could explain the response of spinal progenitors expressing ASCL1 or ATOH1 to altered canonical BMP activity, it would not explain the relative insensitivity of the progenitors expressing NEUROG1, NEUROG2 or PTF1a. Therefore, we tested the veracity of these functional relationships to identify the basis of this differential response (*Figure 1D*).

## ID2 acts downstream of the canonical BMP pathway to differentially regulate the generation of spinal neurons derived from progenitors expressing ASCL1/ATOH1 or NEUROG1/NEUROG2/PTF1a

To test whether ID proteins act downstream of the canonical BMP pathway during spinal neurogenesis, we focused on ID2 (*Figure 2A*), not least because canonical BMP signalling is necessary and sufficient to promote *cId2* expression in the developing spinal cord (*Figure 2—figure supplement 1* and *Le Dréau et al., 2014*). Moreover, the pattern of *cId2* expression closely overlaps that described for the canonical BMP activity during spinal cord development (*Le Dréau et al., 2012*; *Le Dréau et al., 2014*). At early stages during neural patterning, *cId2* expression is restricted to the dorsal region of the developing spinal cord (*Figure 2B*). Later on during neurogenesis this pattern spreads ventrally throughout the D-V axis, showing expression within all ventral progenitor domains except the pMN domain (*Figure 2C–D* and *Figure 2—figure supplement 2*). Inhibition of endogenous ID2 activity was triggered by *in ovo* electroporation of a sh-RNA specifically targeting chick *Id2* transcripts (sh-Id2, *Figure 2—figure supplement 3A–E*). This ID2 inhibition caused premature cell-autonomous differentiation at 48 hpe similar to that provoked by inhibiting SMAD1/5 (*Figure 2E–K* and *Le Dréau et al., 2014*). Conversely, overexpression of a murine ID2 construct reduced the proportion of electroporated cells that differentiated into neurons (EP$^+$;HuC/D$^+$ cells, *Figure 2H,K* and *Figure 2—figure supplement 3F,G*). ID2 overexpression could also partially impede the premature differentiation caused by both sh-Id2 and sh-Smad5 (*Figure 2I–K*). Similar results were obtained when measuring the activity of the pTubb3:luc reporter 24 hpe (*Figure 2L*).

We next analysed the consequences of ID2 inhibition on the generation of the different subtypes of spinal neurons and detected a significant dose-dependent reduction in the generation of many neuronal subtypes (*Figure 2M,N*). The overall phenotype caused by ID2 inhibition was comparable to that triggered by inhibiting BMP7, SMAD1 or SMAD5: the neuronal subtypes deriving from spinal progenitors expressing either ATOH1 or ASCL1 alone were globally more sensitive to ID2 inhibition than those derived from progenitors expressing NEUROG1, NEUROG2 or PTF1a (*Figure 2O*). Together, these results suggest that ID2 acts downstream of the canonical BMP pathway in spinal neurogenesis and that it regulates distinctly the generation of spinal neurons derived from progenitors expressing ASCL1/ATOH1 and NEUROG1/NEUROG2.

## ID2 and E proteins counterbalance each other's activity during spinal neurogenesis

We wondered whether ID2 contributes to spinal neurogenesis by sequestering E proteins (*Figure 3A*). Thus, we analysed the expression of these class I HLH genes during spinal neurogenesis. Transcripts from the *cTcf3/E2A* gene, which encodes the E12 or E47 alternative splice isoforms (*Murre et al., 1989*), were readily detected in the ventricular zone throughout the dorsal-ventral axis of the developing spinal cord, with apparently no domain-specific pattern (*Figure 3B* and *Holmberg et al., 2008*). Transcripts from the chicken HEB orthologue *cTcf12* were detected in the transition zone, following a dorsal-to-ventral gradient (*Figure 3C*). Previous studies reported that *E2-2* transcripts were barely detected in the developing murine spinal cord (*Sobrado et al., 2009*).

The overexpression of E47 or TCF12 both produced a significant increase in the proportion of EP$^+$;HuC/D$^+$ cells (*Figure 3D–F,J*), a phenotype that was reverted by the concomitant electroporation of ID2 (*Figure 3D–J*). To inhibit the endogenous activity of E proteins, we took advantage of an E47 construct carrying mutations in its basic domain (E47bm) and that acts in a dominant-negative manner over E proteins in vivo (*Zhuang et al., 1998*). Electroporation of E47bm inhibited neuronal differentiation in a cell autonomous manner, and it fully compensated for the premature differentiation caused by both E47 and TCF12 (*Figure 3—figure supplement 1*). This E47bm construct also rescued to a large extent the premature differentiation triggered by sh-Id2 (*Figure 3K–O*). Together, these results appear to confirm that the role played by ID2 during spinal neurogenesis depends on its ability to sequester E proteins.

## E47 modulates in opposite ways the neurogenic abilities of ASCL1/ATOH1 and NEUROG1/NEUROG2 during spinal neurogenesis

The results we obtained so far suggested that E proteins themselves might co-operate differently with the distinct proneural proteins during spinal neurogenesis (*Figure 4A*). To test this hypothesis,

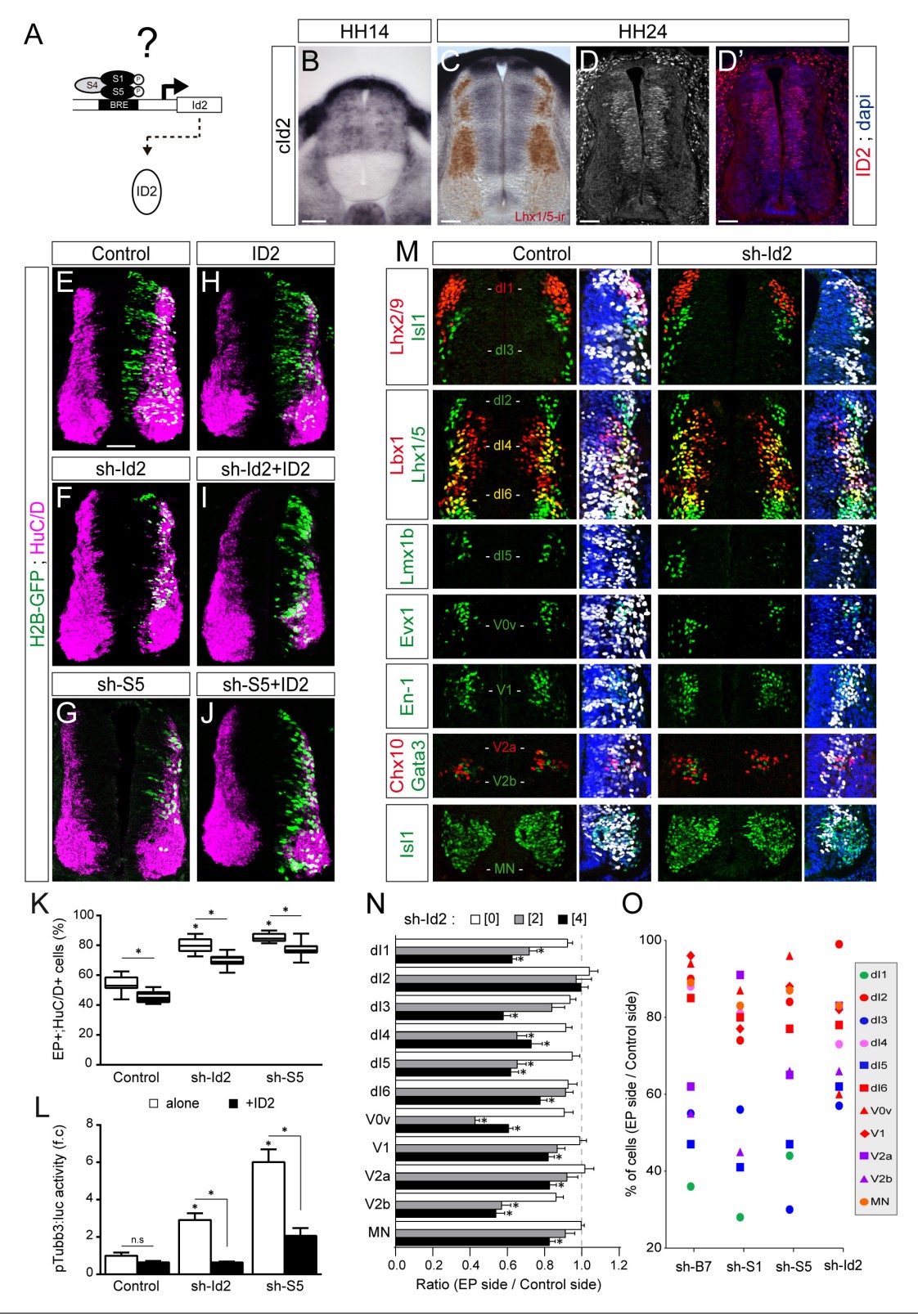

**Figure 2.** ID2 acts downstream of the canonical BMP pathway and it differentially regulates the generation of spinal neurons derived from progenitors expressing ASCL1/ATOH1 or NEUROG1/NEUROG2/PTF1a. (**A**) Hypothesis: ID2 mediates the canonical BMP activity during spinal neurogenesis. (**B, C**) Detection of *cId2* transcripts by in situ hybridization in transverse spinal sections at stages HH14 (**B**) and HH24 (**C**). Lhx1/5 immunoreactivity (brown) was detected a posteriori (**C**). (**D–D'**) Endogenous cID2 immunoreactivity and DAPI staining at stage HH24. (**E–J**) Transverse spinal cord sections of

*Figure 2 continued on next page*

*Figure 2 continued*

electroporated cells (H2B-GFP+) that differentiated into neurons (HuC/D+) 48 hpe with: a control plasmid (**E**), plasmids producing sh-RNAs against *cId2* (sh-Id2, (**F**) or *cSmad5* (sh-S5, (**G**), a murine ID2 construct (**H**), and its combination with sh-Id2 (**I**) or sh-S5 (**J**). (**K**) Box-and-whisker plots obtained from n = 7–16 embryos; one-way ANOVA + Tukey's test; *p<0.05. (**L**) Activity of the pTubb3:luc reporter quantified 24 hpe in the conditions cited above, expressed as the mean fold change ± sem relative to the control, obtained from n = 8–9 embryos; one-way ANOVA + Tukey's test; *p<0.05. (**M**) Representative images of the spinal neuron subtypes (identified with the combinations of the markers indicated) generated 48 hpe with control or sh-Id2. (**N**) Mean ratios ± sem or (**O**) dot-plots comparing the mean number of neurons on the electroporated and contralateral sides, obtained from n = 8–11 embryos; one-way ANOVA + Tukey's test; *p<0.05. Scale bars, 50 μM.

DOI: https://doi.org/10.7554/eLife.37267.005

The following figure supplements are available for figure 2:

**Figure supplement 1.** Regulation of Id2 expression by the canonical BMP pathway.

DOI: https://doi.org/10.7554/eLife.37267.006

**Figure supplement 2.** cId2 expression during spinal neurogenesis.

DOI: https://doi.org/10.7554/eLife.37267.007

**Figure supplement 3.** Modulation of ID2 activity in vivo.

DOI: https://doi.org/10.7554/eLife.37267.008

we first analyzed the consequences of expressing the E47bm mutant on the generation of spinal neuron subtypes. There was a marked reduction ($\geq$50%) in the generation of Lhx2/9$^+$ (dI1) and Tlx3$^+$ (dI3/dI5) interneurons, which derive respectively from progenitors expressing ATOH1 and ASCL1 alone (*Figure 4B,C,F*). By contrast, electroporation of E47bm affected to a lesser extent (<25%) the generation of Lhx1/5$^+$ interneurons (dI2/dI4/dI6-V1) or Isl1$^+$ motor neurons deriving from progenitors expressing NEUROG1 alone (dP2, dP6-V1), PTF1a (dP4) or NEUROG2 (pMN, *Figure 4D–F*). Alternatively, we used another dominant-negative construct of E47: E47Δnls-RFP, inserted in a plasmid with low electroporation efficiency (see the Materials and methods section). This version of E47 fused to RFP is deleted from its nuclear localization signals and thereby impairs the nuclear import of E47, hence its transcriptional activity (*Mehmood et al., 2009*). As previously reported in vitro (*Mehmood et al., 2009*), the subcellular localization of this E47Δnls-RFP mutant after *in ovo* electroporation was mostly cytoplasmic (*Figure 4G–H'*). As seen with E47bm, this E47Δnls-RFP mutant also impaired neuronal differentiation cell-autonomously (*Figure 4G–I*). We analyzed the consequences of E47Δnls-RFP electroporation on the generation of spinal neuron subtypes by quantifying the proportions of differentiated electroporated cells (by focusing on the RFP+ cells that were HuC/D+ or Sox2-) that express distinct neuronal subtype markers (*Figure 4J–R*). Compared to a control plasmid, electroporation of the E47Δnls-RFP mutant reduced about half the proportions of differentiated electroporated cells that express Lhx2/9 or Tlx3 (*Figure 4J–M and R*), indicating that inhibiting E47 activity hindered the differentiation of spinal progenitors expressing ATOH1 or ASCL1 alone. By contrast, progenitors electroporated with the E47Δnls-RFP mutant efficiently differentiated into Lhx1/5$^+$ interneurons or Isl1$^+$ motor neurons (*Figure 4N–R*). Of note, we could even observe Lhx1/5$^+$ or Isl1$^+$ electroporated cells within the ventricular zone (stars in *Figure 4O–O', Q–Q'*), indicative of a premature differentiation of NEUROG1/2/PTF1a-expressing progenitors when E47 activity is impaired. Hence, ATOH1 and ASCL1 appear to be much more dependent on the activity of E proteins to promote appropriate neuronal differentiation than are NEUROG1, NEUROG2 and PTF1a.

Next, we evaluated how E47 gain-of-function modulates the neuronal differentiation induced when ASCL1, ATOH1, NEUROG1 or NEUROG2 are overexpressed (*Figure 5A–H*). From 24 hpe onwards, all four proneural bHLH proteins caused premature differentiation in a cell-autonomous and concentration-dependent manner (*Figure 5—figure supplement 1A–C*). Based on these data, we decided to test how E47 addition would alter the phenotypes caused by sub-optimal concentrations of these four proneural proteins. The addition of E47 accentuated the mild increase in neuronal differentiation provoked by ASCL1 at 24 hpe, and more significantly at 48 hpe (*Figure 4A–B'*). Accordingly, E47 provoked a significant reduction in the average number of electroporated cells generated 48 hpe of ASCL1 (*Figure 4I*). A similar tendency, albeit less pronounced, was observed when E47 was combined with ATOH1, especially in terms of the reduced average number of EP$^+$ cells generated 48 hpe (*Figure 4C–D',I*). Addition of E47 had the opposite effect when combined with NEUROG1 or NEUROG2: it significantly reduced the proportion of EP$^+$;HuC/D$^+$ cells obtained at 24 hpe and consequently increased the final numbers of EP$^+$ cells observed at 48 hpe (*Figure 5E–*

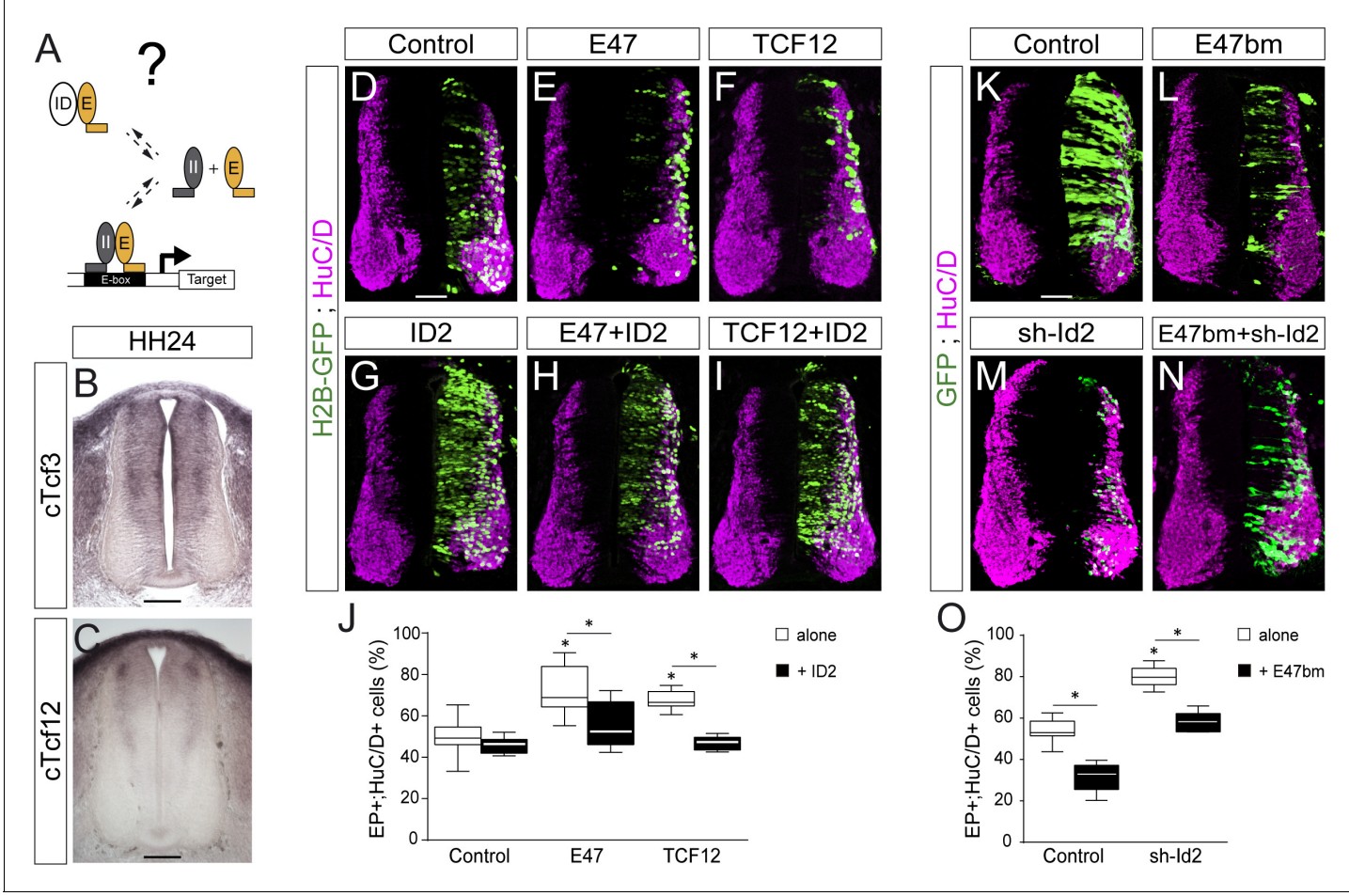

**Figure 3.** ID2 and E proteins counteract each other's activity during spinal neurogenesis. (**A**) Hypothesis: ID2 sequesters E proteins during spinal neurogenesis. (**B, C**) Detection of *cTcf3/cE2a* (**B**) and *cTfc12* (**C**) transcripts by in situ hybridization in transverse spinal cord sections at stage HH24. (**D**–**O**) Transverse spinal cord sections of electroporated cells (GFP+ or H2B-GFP+) that differentiated into neurons (HuC/D+) 48 hpe with: a control (**D**), E47 (**E**), TCF12 (**F**), ID2 (**G**) or combinations of these (**H, I**); a control (**K**), E47bm (**L**), sh-Id2 (**M**) or their combination (**N**). (**J, O**) Box-and-whisker plots obtained from n = 7–16 (**J**) and n = 9–16 (**O**) embryos; one-way ANOVA + Tukey's test; *p<0.05. Scale bars, 50 µM. .
DOI: https://doi.org/10.7554/eLife.37267.009

The following figure supplement is available for figure 3:

**Figure supplement 1.** E47bm rescues the premature neuronal differentiation caused by both E47 and TCF12.
DOI: https://doi.org/10.7554/eLife.37267.010

*I*). These results suggested that E47 differentially regulates the ability of ASCL1/ATOH1 and NEUROG1/NEUROG2 to promote cell cycle exit.

To assess cell cycle exit, a fluorescent cytoplasmic-retention dye that is only diluted on cell division was added at the time of electroporation and its mean fluorescence intensity was measured in FACS-sorted electroporated (GFP+) cells 48 hr later (*Figure 5J*). This assay demonstrated that E47 itself increased the mean Violet intensity, and further enhanced the mild increase caused by ASCL1 (*Figure 5K*), indicating that E47 facilitates ASCL1's ability to promote cell cycle exit. E47 had an opposite effect when combined with NEUROG1, significantly reducing the strong increase in Violet intensity caused by NEUROG1 (*Figure 5K*), thereby confirming that E47 restricts NEUROG1's ability to promote cell cycle exit.

We next studied how E47 influences the respective abilities of ASCL1 and NEUROG1 to regulate the balance between the three different modes of division that spinal progenitors can undergo during neurogenesis: symmetric proliferative divisions (PP), asymmetric divisions (PN), and symmetric neurogenic divisions (NN) (*Saade et al., 2013*; *Le Dréau et al., 2014*). To this end, we took

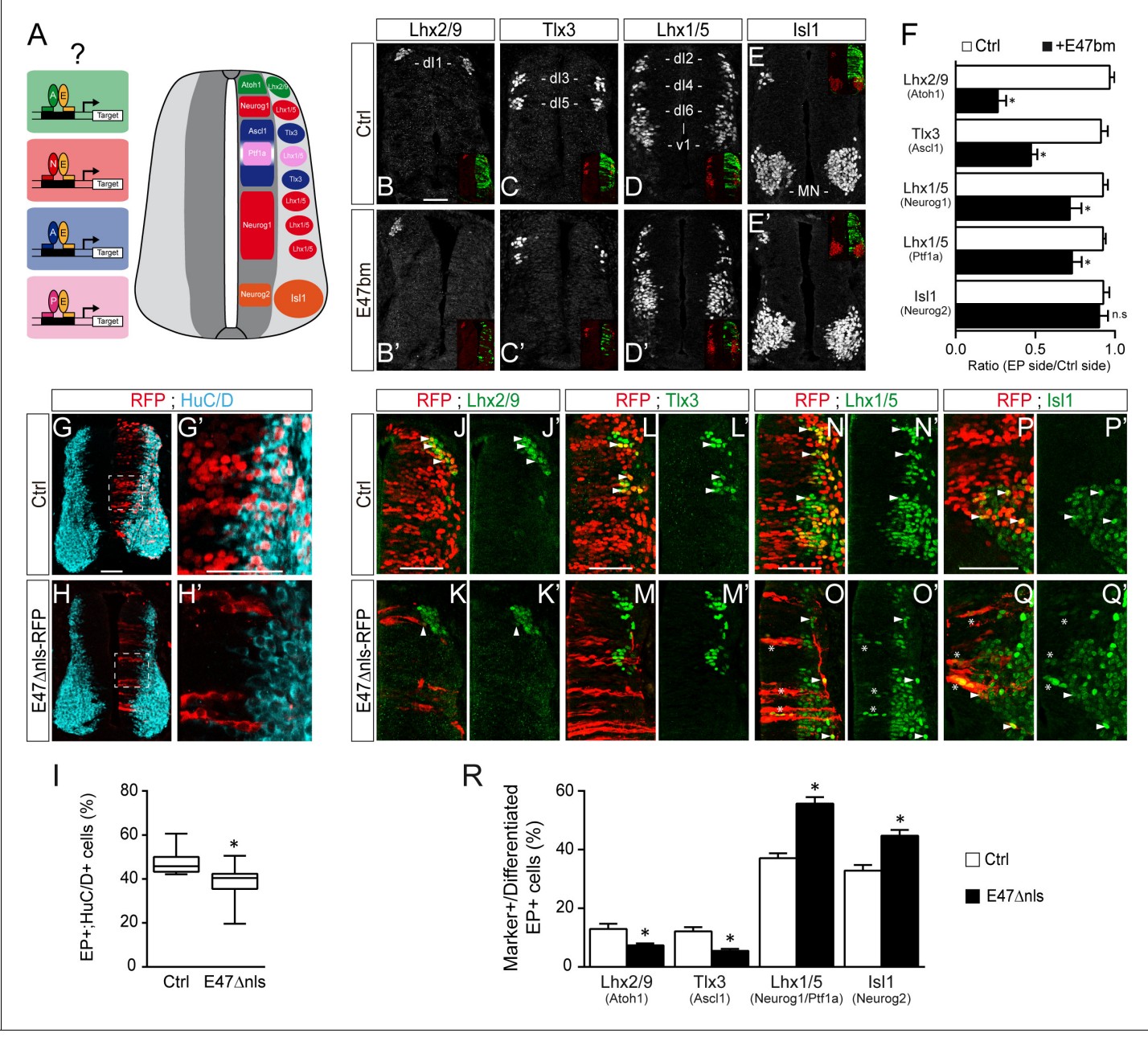

**Figure 4.** E47 activity is differentially required for the generation of spinal neurons deriving from progenitors expressing ASCL1/ATOH1 and NEUROG1/NEUROG2/PTF1a. (A) Hypothesis: E proteins co-operate differently with the distinct proneural proteins during spinal neurogenesis. (B–E) Representative images of spinal neurons expressing Lhx2/9 (dl1, (B–B'), Tlx3 (dl3/dl5, (C–C'), Lhx1/5 (dl2/dl4/dl6-V1, (D–D') or Isl1 (MN, (E–E'), 48 hpe with a control (B–E) or E47bm (B'–E'). (F) Mean ratios ± sem of neuron numbers on the electroporated side relative to the contralateral side, obtained from n = 8–13 embryos; two-sided unpaired t-test; *p<0.05. (G–H') Transverse spinal cord sections of electroporated cells (RFP+) that differentiated into neurons (HuC/D+) 48 hpe with a control plasmid (G) or a plasmid expressing an E47Dnls-RFP fusion construct (H). (I) Box-and-whisker plots obtained from n = 9–12 embryos; Mann-Whitney's test; *p<0.05. (J–Q') Transverse spinal cord sections showing the electroporated cells (RFP+) that differentiated into Lhx2/9+ (J–K'), Tlx3+ (L–M') or Lhx1/5+ (N–O') interneurons or Isl1+ motoneurons (P–Q') 48 hpe with a control plasmid (J,L,N,P) or a plasmid expressing an E47Dnls-RFP fusion construct (K,M,O,Q). Examples of differentiated RFP+;marker+ cells are highlighted by arrowheads, or by stars for the electroporated cells that were found prematurely differentiated in the ventricular zone. (R) Proportions of differentiated electroporated cells (RFP+;HuC/D+ or RFP+;Sox2-) that express the distinct neuronal subtype markers mentioned above, obtained from n = 10–12 embryos; Mann‗Whitney's non-parametric test for Lhx2/9 or two-sided unpaired t-test for the three other markers; *p<0.05. Scale bars, 50 μM.
DOI: https://doi.org/10.7554/eLife.37267.011

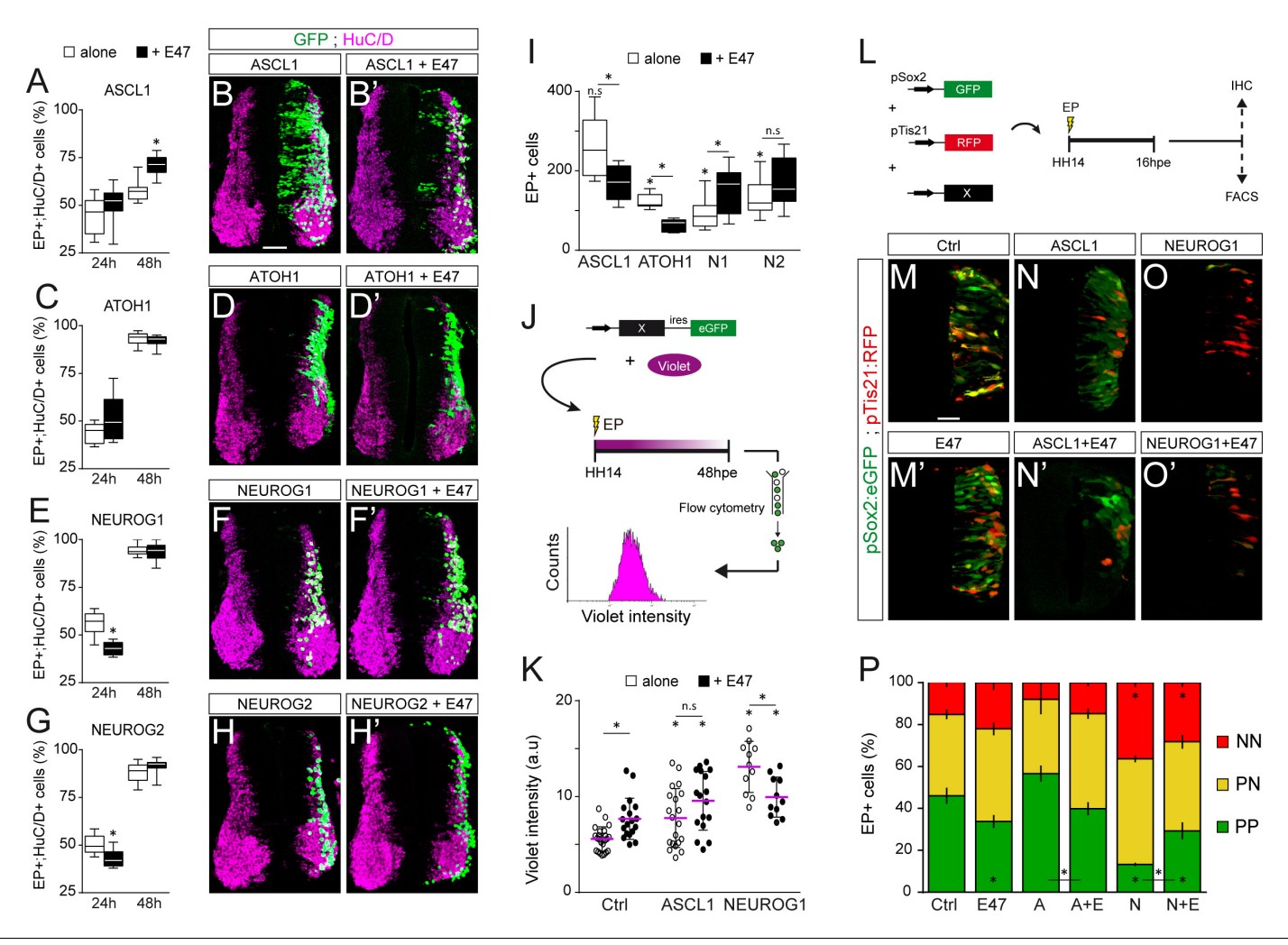

**Figure 5.** E47 modulates in opposite ways the neurogenic abilities of ASCL1/ATOH1 and NEUROG1/NEUROG2 during spinal neurogenesis. (A–H')
Transverse spinal cord sections of electroporated cells (GFP +or H2B-GFP+) that differentiated into neurons (HuC/D+) 24 and 48 hpe with ASCL1 (A–B),
ATOH1 (C–D), NEUROG1 (E–F) or NEUROG2 (G–H) alone (white) or together with E47 (black, A'-H'). Box-and-whisker plots obtained from n = 6–9 (A),
6–8 (C), 6–14 (E) and 7–12 (G) embryos; two-way ANOVA + Sidak's test; *p<0.05. (I) Mean number ±sem of electroporated cells quantified 48 hpe with the
proneural proteins on their own (white) or together with E47 (black), obtained from 6 to 14 embryos; two-sided unpaired t-test; *p<0.05. (J) Cell
cycle exit assay. (K) Mean Violet fluorescence intensity measured 48 hpe with a control, ASCL1 and NEUROG1 on their own (white) or together with E47
(black). The individual values (dots, n = 11–23 embryos) and the mean (bars) are shown; one-way ANOVA + Tukey's test and two-way ANOVA + Sidak's
test; *p<0.05. (L) Assessment of the modes of division of spinal progenitors. (M–O) Transverse spinal cord sections showing the activity of the pSox2:
GFP and pTis21:RFP reporters at 16 hpe, when electroporated in combination with control, ASCL1 or NEUROG1 on their own (M–O) or together with
E47 (M'-O'). (P) Mean proportion ±sem of cells identified as pSox2+/pTis21- (PP), pSox2 +/pTis21+ (PN) or pSox2-/pTis21+ (NN) when quantified by
FACS, obtained from n = 6–10 pools of embryos; two-way ANOVA + Tukey's test; *p<0.05. Scale bars, 50 μM. .

DOI: https://doi.org/10.7554/eLife.37267.012

The following figure supplement is available for figure 5:

**Figure supplement 1.** Effects of E47 and proneural proteins on spinal neuronal differentiation.

DOI: https://doi.org/10.7554/eLife.37267.013

advantage of the pSox2:eGFP and pTis21:RFP reporters that are specifically active during progenitor-generating (PP +PN) and neuron-generating (PN +NN) divisions, respectively (*Saade et al., 2013*). The effects of E47, ASCL1 and NEUROG1 on the reporters' activities were assayed 16 hpe by immunohistochemistry or quantified by FACS (*Figure 5L*). E47 caused a significant decrease in the proportion of pSox2:eGFP+;pTis21:RFP- (PP) cells and a reciprocal increase in the proportion of pTis21:RFP+ (PN +NN) neurogenic divisions relative to the controls (*Figure 5M,M',P*). While we did

not detect any significant change in the proportions of PP, PN and NN cells in response to ASCL1 alone at this concentration, we did observe an increase in neurogenic divisions at the expense of proliferative divisions when ASCL1 was combined with E47 (*Figure 5N,N',P*). Conversely, E47 significantly restrained NEUROG1's ability to trigger neurogenic divisions at the expense of PP divisions (*Figure 5O–P*). Assessing the activity of the pSox2:luc reporter confirmed these results, further showing that E47 facilitates the ability of both ASCL1 and ATOH1 to repress pSox2 activity, whereas it restricts the repressive effects of both NEUROG1 and NEUROG2 (*Figure 5—figure supplement 1D*). Together, these results revealed that E47 co-operates distinctly with ASCL1/ATOH1 and NEUROG1/NEUROG2 to fine-tune neurogenic divisions during spinal neurogenesis.

## E47 modulates the transcriptional activities of ASCL1 and NEUROG1 in an E-box dependent manner and through physical interactions

To identify the molecular mechanisms underlying the different outcomes caused by E proteins' co-operation with the distinct proneural proteins, we focused on the interaction of E47 with ASCL1 and NEUROG1. A DNA-binding deficient version of NEUROG1 (NEUROG1-AQ, *Sun et al., 2001*), was unable to transactivate the NEUROG-responsive pNeuroD:luc reporter or to promote neuronal differentiation (*Figure 6—figure supplement 1*). Hence, the ability of NEUROG1 to trigger neuronal differentiation during spinal neurogenesis depends on its transcriptional activity, as previously reported for ASCL1 and ATOH1 (*Nakada et al., 2004*).

Genome-wide ChIP-seq studies have established that the preferential E-box motifs bound by ASCL1, E47 and NEUROG1 correspond respectively to CAGCTG (*Castro et al., 2011*; *Borromeo et al., 2014*), CAGSTG (where S stands for C or G: *Lin et al., 2010*; *Pfurr et al., 2017*) and CADATG (where D stands for A, G or T: *Seo et al., 2007*; *Madelaine and Blader, 2011*). In light of these intrinsic preferences, we tested how E47 modulates the abilities of ASCL1 and NEUROG1 to bind to DNA and activate transcription in different E-box contexts (*Figure 6A* and *Figure 6—figure supplement 2A*). E47 acted in synergy with both ASCL1 and NEUROG1 to drive transcription of the pkE7:luc reporter under the control of 7 CAGGTG repeats (*Figure 6B,C*). By contrast, E47 and ASCL1 only weakly transactivated the pNeuroD:luc reporter, the promoter of which contains 9 CADATG E-boxes and 1 CAGGTG box (*Figure 6D*). A similar result was obtained with a version of the pNeuroD:luc reporter in which the single CAGGTG motif was destroyed by mutagenesis (*Figure 6—figure supplement 2B*), reinforcing the idea that both E47 and ASCL1 preferentially bind to CAGSTG sequences. Intriguingly, E47 markedly reduced the ability of NEUROG1 to enhance the activity of both pNeuroD:luc and its mutated version (*Figure 6E* and *Figure 6—figure supplement 2C*). Of note, TCF12 was also able to enhance ASCL1-dependent pKE7:luc activity while inhibiting NEUROG1's ability to induce pNeuroD:luc activity, though with milder capacities than E47 (*Figure 6—figure supplement 2D,E*). To further define whether the way E47 modulates ASCL1 and NEUROG1's transcriptional activity directly depends on the E-box content, we used two additional reporters: pDll1-M:luc and pDll1-N:luc. These are based on conserved regulatory elements found in the promoter of the *Delta-like1* gene and have been described to respectively respond to ASCL1 and NEUROG2 (*Beckers et al., 2000*; *Castro et al., 2006*). When combined with the pDll1-M:luc reporter containing 3 CAGSTG +1 CADATG motifs, addition of E47 to ASCL1 had only an additive effect (*Figure 6—figure supplement 2F*), compared to the synergistic effect observed on pKE7:luc activity (*Figure 6B*). When combined with the pDll1-N:luc reporter containing 1 CAGSTG +3 CADATG motifs, addition of E47 still inhibited the activity induced by NEUROG1 (*Figure 6—figure supplement 2G*), but less potently than when combined with the pNeuroD:luc reporter (*Figure 6E*). In vitro ChIP assays further demonstrated that E47 can bind to and enhance ASCL1 binding at the 7 CAGGTG-containing promoter region of the pkE7:luc reporter (*Figure 6F*), consistent with the notion that their heterodimerization is required for optimal binding and subsequent transcriptional activation. By contrast, E47 caused a significant reduction in the amount of NEUROG1 bound to the promoter region of the pNeuroD:luc reporter (*Figure 6G*). The fact that E47 itself bound to this promoter region suggested that E47 and NEUROG1 compete for binding to CADATG motifs (*Figure 6G*), although E47 cannot transactivate them as potently as NEUROG1 (*Figure 6E*). Together, these results revealed that E47 acts in synergy with both ASCL1 and NEUROG1 when binding to its own optimal E-box (CAGSTG), while it somehow impedes NEUROG1 from binding to CADATG motifs.

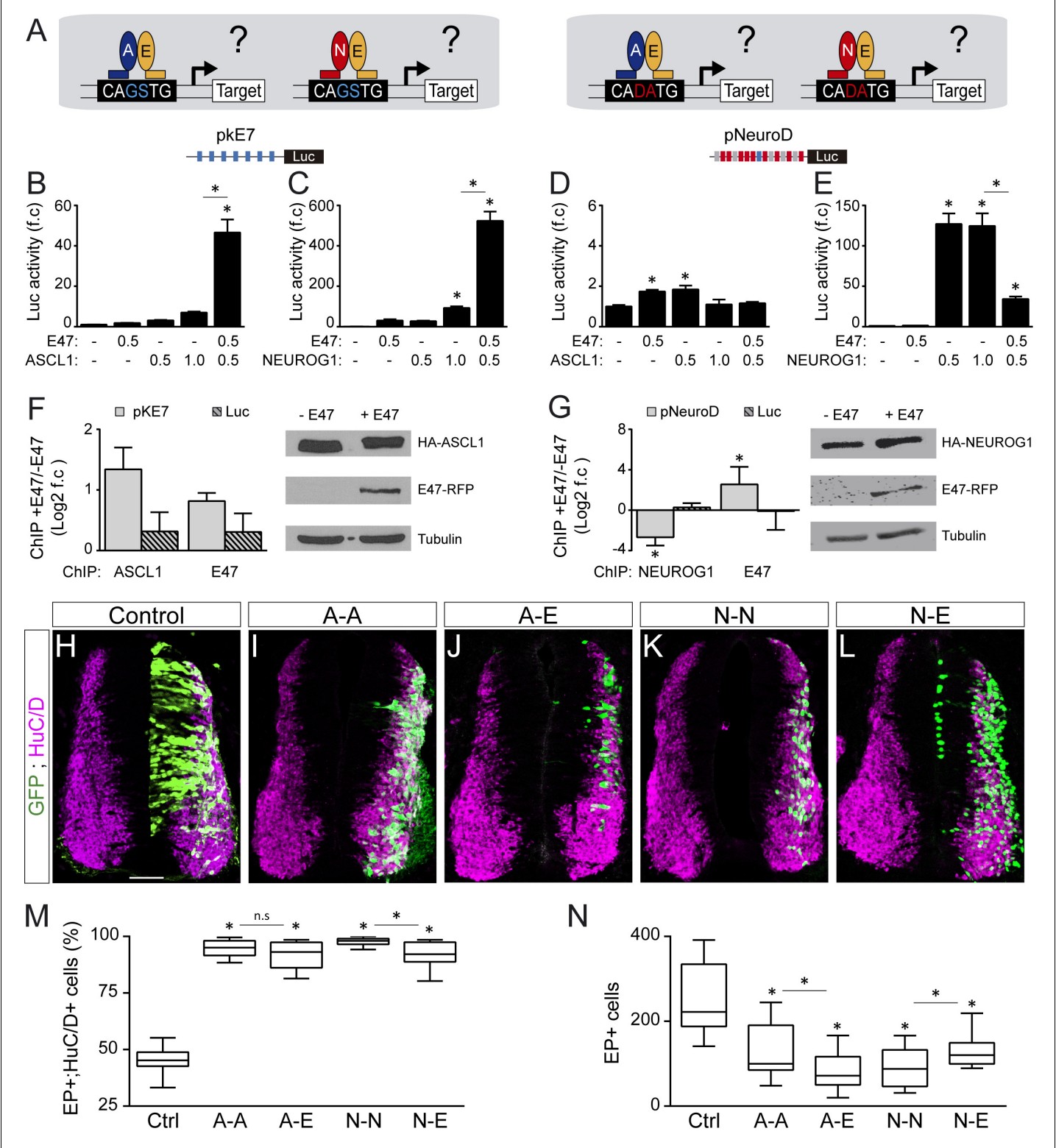

**Figure 6.** E47 modulates the transcriptional activities of ASCL1 and NEUROG1 in an E-box-dependent manner and through physical interactions. (**A**) Hypothesis: E proteins modulate the activity of the proneural proteins differently depending on the E-box context. (**B–E**) Activity of the pkE7 (**B**, **C**) and pNeuroD (**D**, **E**) luciferase reporters measured 24 hpe with a control, E47 and ASCL1 (**B**, **D**) or NEUROG1 (**C**, **E**), expressed as the mean fold change ±sem relative to the control, obtained from n = 8 embryos; one-way ANOVA + Tukey's test. (**F–G**) ChIP assays performed on the pkE7 (**F**) or pNeuroD (**G**) promoter regions (light grey), or luciferase ORF (Luc, striped grey), in HEK293 cells 24 hr after transfection with HA-ASCL1 (**F**) or HA-
*Figure 6 continued on next page*

*Figure 6 continued*

NEUROG1 (**G**) on their own or together with E47-RFP, expressed as Log2 values of the mean fold change ±sem in DNA binding measured in the presence of E47 relative to absence of E47, obtained from n = 3 (**F**) or n = 5 (**G**) experiments; two-sided one sample t-test. The HA-ASCL1, HA-NEUROG1 and E47-RFP proteins probed in western blots, with Tubulin-beta as a transfection control. (**H–L**) Transverse spinal cord sections of electroporated cells (GFP+) that differentiated into neurons (HuC/D+) 48 hpe with a control (**H**), ASCL1 or NEUROG1 homodimer (A-A, I; **N–N, K**), or ASCL1-E47 or NEUROG1-E47 heterodimers (A-E, J; **N–E, L**). (**M**) Box-and-whisker plots obtained from n = 12–15 embryos; one-way ANOVA + Tukey's test. (**N**) Mean number of electroporated cells (GFP+) generated 48 hpe in the conditions cited above, calculated from n = 11–14 embryos; one-way ANOVA + Tukey's test. *p<0.05. Scale bars, 50 μM. .
DOI: https://doi.org/10.7554/eLife.37267.014

The following figure supplements are available for figure 6:

**Figure supplement 1.** The ability of NEUROG1 to induce spinal neuronal differentiation depends on its DNA-binding.
DOI: https://doi.org/10.7554/eLife.37267.015
**Figure supplement 2.** E-box dependent activity of E47, TCF12, ASCL1 and NEUROG1 during spinal neurogenesis.
DOI: https://doi.org/10.7554/eLife.37267.016
**Figure supplement 3.** Characterization of the tethered constructs of bHLH dimers.
DOI: https://doi.org/10.7554/eLife.37267.017

To assess whether these E-box-dependent modulations of proneural proteins' transcriptional activity by E47 indeed depend on physical interactions, we first prevented E47 from interacting with ASCL1 or NEUROG1 by adding ID2. As expected, addition of ID2 partially rescued both the synergistic effect of E47 on pKE7:luc activity when combined with ASCL1 and the inhibitory effect of E47 on NEUROG1-induced pNeuroD:luc activity (*Figure 6—figure supplement 2H,I*). Secondly, we compared the activity of tethered constructs that were designed to produce homodimers of ASCL1 (A-A) and NEUROG1 (N-N), or heterodimers with E47 (A-E, N-E: *Figure 6—figure supplement 3A–C*). Consistent with the results obtained with monomers, A-E heterodimers were significantly more potent than A-A homodimers in driving pkE7:luc activity (*Figure 6—figure supplement 3D*), while N-N homodimers transactivated pNeuroD:luc much more strongly than N-E heterodimers (*Figure 6—figure supplement 3E*). A-A and A-E promoted similar neuronal differentiation 48 hpe (*Figure 6H–J,M*), but the average number of EP⁺ cells obtained after A-E electroporation was significantly lower than after A-A electroporation (*Figure 6N*), suggesting that A-E promotes early neurogenic divisions more potently than A-A. This idea was supported by the ability of A-E to repress pSox2:luc activity at 20 hpe, unlike A-A (*Figure 6—figure supplement 3F*). As for NEUROG1, N-N was significantly more potent than N-E at promoting neuronal differentiation (*Figure 6K–M*), at reducing the average number of EP⁺ cells generated 48 hpe (*Figure 6N*) and at repressing pSox2:luc activity (*Figure 6—figure supplement 3F*). Thus, the tethered constructs performed like the monomers (*Figure 5*), supporting the conclusion that E47 facilitates the ability of ASCL1 and restrains that of NEUROG1 to trigger neurogenic divisions during spinal neurogenesis.

## E47 modulates in opposite ways the neurogenic abilities of ASCL1 and NEUROG1/NEUROG2 during corticogenesis

We were interested to determine if this differential co-operation of E47 with the distinct proneural proteins could be extended to other regions of the developing CNS. We tested this hypothesis in the developing cerebral cortex, as NEUROG1/2 and ASCL1 all contribute to neurogenesis in this region in mammals (*Huang et al., 2014*). The development of the cerebral cortex in birds actually shows unexpected similarities to mammalian corticogenesis, including the conservation of its temporal sequence of neurogenesis (*Dugas-Ford et al., 2012*; *Suzuki et al., 2012*). As in mammals, corticogenesis in the chick embryo originates from a region of the dorsal pallium expressing PAX6 (*Figure 7A,B* and *Suzuki et al., 2012*). From E3 to E5, an early phase of corticogenesis produces the first SOX2⁻;HuC/D⁺ cortical neurons, which are generated specifically from PAX6⁺;TBR2⁻ radial glia-like progenitors that divide at the apical surface, as in mammals (*Figure 7C–D*). Cortical TBR2⁺ progenitors that divide basally, similar to mammalian intermediate progenitor cells, appear at around E5 (*Figure 7D–D''*). The cortical neurons produced during this early phase express TBR1 (*Figure 7—figure supplement 1A–A''*), as well as other markers typically expressed by mammalian deep-layer neurons (*Dugas-Ford et al., 2012*; *Suzuki et al., 2012*). From E3 to E5, *cTcf3/cE2A* transcripts were detected throughout the whole D-V axis of the developing telencephalon, with apparently no

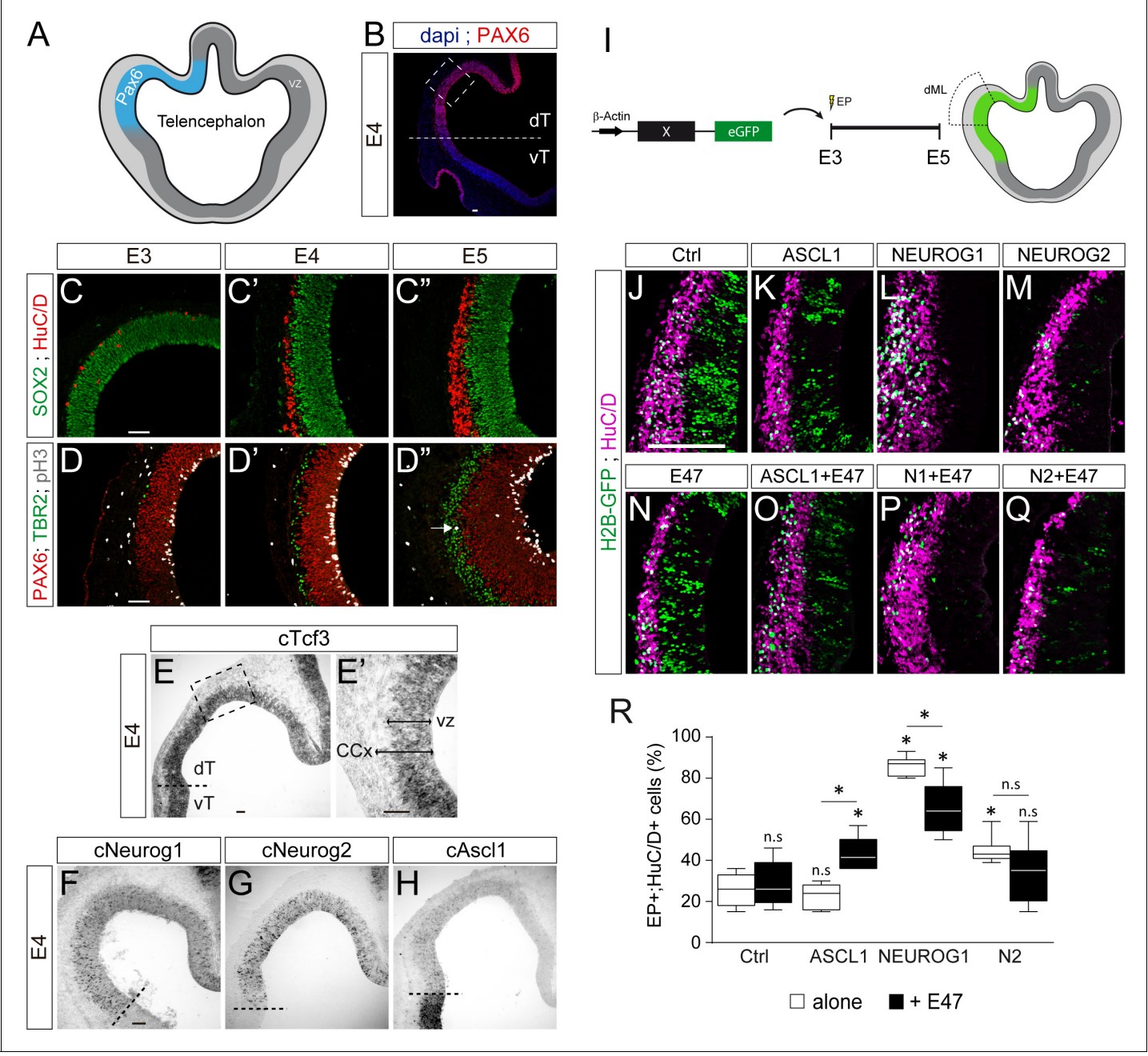

**Figure 7.** E47 modulates in opposite ways the neurogenic abilities of ASCL1 and NEUROG1/NEUROG2 during corticogenesis. (**A**) Scheme of the embryonic chick telencephalon at early stages of neurogenesis. (**B–D**) Coronal telencephalic sections showing PAX6 immunoreactivity and the cell nuclei (DAPI) at low magnification at E4 (**B**), cortical progenitors and differentiating neurons (SOX2 +and HuC/D+, (**C**), apical progenitors (PAX6+;TBR2-, (**D**) and mitotic basal progenitors (TBR2+;pH3+, arrow in D'') at E3 (**C, D**), E4 (**C', D'**) and E5 (**C'', D''**). (**E–H**) Detection of *cTcf3/cE2a* (**E**), *cNeurog1* (**F**), *cNeurog2* (**G**) and *cAscl1* (**H**) transcripts by in situ hybridization at E4. (**I**) *In ovo* electroporation of the chick telencephalon. (**J–Q**) Coronal telencephalic sections of electroporated cells (GFP+) that differentiated into neurons (HuC/D+) 48 hpe with a control (**J**), ASCL1 (**K**), NEUROG1 (**L**) or NEUROG2 (**M**) on their own or together with E47 (**N–Q**). (**R**) Box-and-whisker plots obtained from n = 5–9 embryos; one-way ANOVA + Tukey's test; *p<0.05. Scale bars, 50 µM. CCx, cerebral cortex; dT/vT, dorsal and ventral telencephalon; dML, dorso-medial-lateral; VZ, ventricular zone.

DOI: https://doi.org/10.7554/eLife.37267.018

The following figure supplement is available for figure 7:

**Figure supplement 1.** Neurogenesis, cTcf3 expression and concentration-dependent effects of proneural proteins during early chick corticogenesis.

DOI: https://doi.org/10.7554/eLife.37267.019

domain-specific pattern (*Figure 7—figure supplement 1B–B″*). *cTcf3/cE2A* transcripts were mainly detected in the ventricular zone formed by cortical progenitors (*Figure 7E–E′*), as previously reported during mouse corticogenesis (*Li et al., 2012*). At E4, *cNeurog1* and *cNeurog2* transcripts were detected in a salt-and-pepper fashion in the cortical PAX6$^+$ region (*Figure 7F,G*), whereas *cAscl1* expression was detected strongly in the sub-pallium and more weakly in the developing cerebral cortex (*Figure 7H*). These expression patterns seen in early chicken embryos are very similar to what is observed in the developing mammalian telencephalon (*Huang et al., 2014*), suggesting the phylogenetic conservation of the functions played by proneural proteins during corticogenesis.

To test how E47 modulates the activity of ASCL1 and NEUROG1/2 in the developing chick cerebral cortex, we electroporated the corresponding proneural constructs in the dorsal telencephalic region *in ovo* at E3 and analysed their effects on neuronal differentiation 2 days later (*Figure 7I*). Both NEUROG1 and NEUROG2 triggered significant neuronal differentiation in the developing cerebral cortex in a cell autonomous and dose-dependent manner, whereas ASCL1 overexpression had only a minor effect per se (*Figure 7J–M,R* and *Figure 7—figure supplement 1C*). E47, which itself had no obvious effect at this concentration (*Figure 7N,R*), significantly increased neuronal differentiation when combined with ASCL1 (*Figure 7O,R*). Conversely, E47 markedly reduced the ability of NEUROG1, and to a lesser extent that of NEUROG2, to promote neuronal differentiation (*Figure 7P–R*). These results suggest that E47 also modulates in opposite ways the neurogenic activities of ASCL1 and NEUROG1/2 in the context of cortical neurogenesis.

## Discussion

Class I HLH/E proteins are generally described as obligatory and permissive co-factors for proneural proteins, which must form heterodimers to become active and regulate transcription (*Wang and Baker, 2015*). The main findings of our study are that the co-operation between E proteins and proneural proteins might be more complex than originally thought. Our results indeed revealed that E proteins can facilitate or restrain the transcriptional activity of the proneural proteins, depending both on their intrinsic DNA-binding preferences and on the E-box content (*Figure 8*).

On the one hand, our results support a revised model whereby E proteins synergize with proneural proteins specifically at CAGSTG E-boxes, the preferential motifs of E proteins (*Lin et al., 2010*; *Pfurr et al., 2017*). Therefore, E proteins facilitate the activity of the proneural proteins that share their preferential binding to CAGSTG motifs, such as ASCL1 and ATOH1 (*Castro et al., 2011*; *Klisch et al., 2011*; *Lai et al., 2011*; *Borromeo et al., 2014*). Inhibiting the activity of E proteins by overexpressing the E47bm or E47Δnls-RFP mutants strongly impaired the generation of interneurons derived from spinal progenitors that express ATOH1 or ASCL1 alone. Conversely, enhancing the expression of E47 reinforced the ability of ATOH1 and more markedly, that of ASCL1 to promote neuronal differentiation. Our results suggest that this results from the capacity of E47 to increase the ability of these proneural proteins to trigger neurogenic divisions at the expense of proliferative ones (*Figure 8*). Such co-operation appears to be particularly crucial in the case of ASCL1, whose overexpression could barely increase neurogenic divisions per se, at least at low concentration. These observations support a growing body of evidence that ASCL1 possesses a mild neurogenic potential. For instance, the broad dP3-dP5 domain of spinal progenitors, in which ASCL1 is expressed alone or in combination with PTF1a, expands at the end of primary neurogenesis before producing large numbers of dILA/B neurons during the second neurogenic wave (*Wildner et al., 2006*; *Borromeo et al., 2014*). Later on, ASCL1 is also involved in promoting oligodendrogenesis in both the developing brain and spinal cord (*Huang et al., 2014*). Moreover, recent studies have reported cell cycle promoting-genes among the targets bound by ASCL1 in the ventral telencephalon and that it also sustains the proliferation of adult neural stem cells (*Castro et al., 2011*; *Urbán et al., 2016*), suggesting that its mild neurogenic ability might actually be required to sustain long-term production of the neural lineages. Whether the ability of ASCL1 to maintain neural progenitor pools is related to its dependence on the availability of E proteins is an intriguing hypothesis that would be worth testing.

On the other hand, our findings demonstrate that E proteins inhibit proneural protein binding to CADATG motifs. In consequence, E proteins restrict the activity of the proneural proteins that preferentially bind to these motifs, such as NEUROG1/2 (*Seo et al., 2007*; *Madelaine and Blader, 2011*). Spinal progenitors electroporated with the E47Δnls-RFP mutant efficiently differentiated into

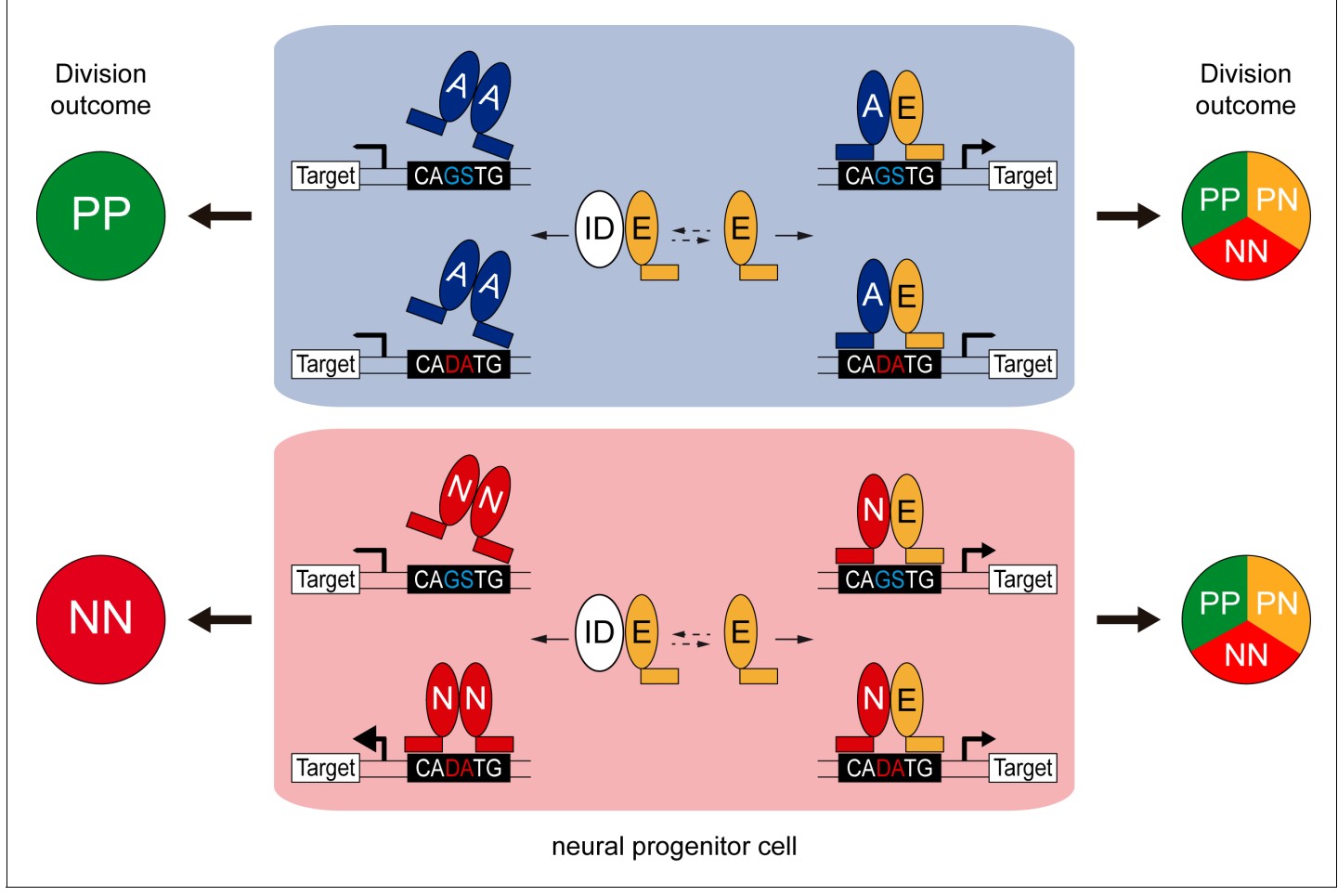

**Figure 8.** Model of the E-box-dependent co-operation of E proteins with proneural proteins. In neural progenitors, ID proteins (ID) physically sequester E proteins (E), thereby regulating their ability to interact with ASCL1 and ATOH1 (A) or NEUROG1/2 (N). When E protein availability is limited, ASCL1/ATOH1 cannot bind optimally to high affinity CAGSTG E-box motifs, resulting in poor regulation of their target genes and favouring symmetric proliferative (PP) divisions and hence, progenitor maintenance. The release of E proteins from IDs allows heterodimerization with ASCL1/ATOH1, resulting in optimal binding to CAGSTG motifs, correct regulation of the target genes and the appropriate increase in neurogenic asymmetric (PN) and self-consuming (NN) divisions. In the absence of E proteins, NEUROG1/2 bind to high affinity CADATG motifs, possibly as homodimers, and regulate the expression of target genes in an exacerbated manner. This deregulation results in excessive neurogenic divisions that cause premature neuronal differentiation and depletion of the progenitor pool. In the presence of E proteins and when N-E heterodimers are formed, the activity of NEUROG1/2 is moderated and the proportions of the different modes of divisions are balanced appropriately to sustain the progenitor population while promoting correct neuronal differentiation.

DOI: https://doi.org/10.7554/eLife.37267.020

Lhx1/5[+] interneurons or Isl1[+] motor neurons. Some of these differentiated cells were even observed within the ventricular zone, evidencing that NEUROG1/2/PTF1a-expressing progenitors are prone to differentiate prematurely when E47 activity is impaired. Since E47 restrains the capacity of NEU-ROG1/2 to promote neuronal differentiation in the context of both spinal neurogenesis and cortico-genesis, this would appear to be a general feature. Depletion of the murine E47 isoform was recently shown to increase the production of both TBR1[+] and SATB2[+] neurons at mid-corticogenesis (*Pfurr et al., 2017*). In fact, the loss of E47 in early cortical progenitors, for which NEUROG2 constitutes the main proneural protein, causes premature neuronal differentiation. This is consistent with our results and model and it contrasts with the block in neuronal differentiation that would be expected if E47 was essential for NEUROG2 activity.

Our results also suggest that NEUROGs do not necessarily need to form heterodimers with E proteins to trigger neuronal differentiation. Indeed, forced NEUROG1 homodimers drive CADATG-

dependent transcription and neuronal differentiation more efficiently than NEUROG1-E47 heterodimers. Similarly, NEUROG2 homodimers better transactivate neuronal differentiation genes than NEUROG2-E47 heterodimers (*Li et al., 2012*), and EMSA experiments suggested the existence of multiple combinations of proneural homo- and heterodimers (*Henke et al., 2009*). The physiological relevance of such proneural homodimers is worthy of further study but to date, our attempts to determine whether NEUROG1 homodimers are formed in vivo during spinal neurogenesis remain inconclusive for technical reasons (data not shown). Nevertheless, the strong capacity of NEUROGs to trigger neurogenic divisions independently of E proteins, including self-consuming NN divisions, correlates well with the fact that neural progenitors expressing NEUROG1/NEUROG2 are usually depleted during the neurogenic phase (*Simmons et al., 2001*; *Kim et al., 2011*). Together, these results support the notion that E proteins are required to dampen the strong capacity of NEUROGs to trigger neurogenic divisions, thereby avoiding the premature depletion of neural progenitor pools (*Figure 8*).

The findings that E proteins modulate in opposite ways the activities of ASCL1/ATOH1 and NEUROG1/2 would also explain why modulating canonical BMP activity affects differently the generation of the distinct neuronal subtypes produced during primary spinal neurogenesis. Inhibiting BMP7 or SMAD1/5 would result in the release of E proteins from their complexes with IDs. In turn, this would facilitate ATOH1 and ASCL1 activity, prematurely increasing the proportion of neurogenic divisions undertaken by the corresponding dP1 and dP3/dP5/p2 progenitors, causing their premature differentiation and exhaustion, and ultimately leading to a production of fewer neurons. As NEUROGs are less dependent on E proteins, the inhibition of canonical BMP signalling only mildly impairs the generation of the neuronal subtypes that derive from progenitors expressing NEUROG1/NEUROG2.

In summary, the results presented here led us to propose that E proteins fine-tune neurogenesis by buffering the activity of the distinct proneural proteins. As such, these data add another layer of sophistication to the molecular mechanisms that regulate the activity of proneural bHLH proteins and hence, neurogenesis.

# Materials and methods

**Key resources table**

| Reagent type (species) or resource | Designation | Source or reference | Identifiers | Additional information |
|---|---|---|---|---|
| Strain, strain background (*Gallus gallus domesticus*) | Chicken fertilized eggs | Granja Gibert | | |
| Cell line (human) | HEK293 | ATCC | RRID:CVCL_0045 | |
| Transfected construct (empty) | pCAGGS:_ires_ GFP | (*Megason and McMahon, 2002*) | | |
| Transfected construct (*Rattus norvegicus*) | pCAGGS:ASCL1 | Other | | Gift from François Guillemot, Francis Crick Institute, London, UK |
| Transfected construct (*Rattus norvegicus*) | pCAGGS:ASCL1-t-E47_ires_GFP | (*Geoffroy et al., 2009*) | | |
| Transfected construct (*Rattus norvegicus*) | pCAGGS:ASCL1-t-ASCL1_ires_GFP | This paper | | See Materials and methods |
| Transfected construct (*Mus musculus*) | pCAGGS:ATOH1_ ires_GFP | This paper | | See Materials and methods |
| Transfected construct (*Mus musculus*) | pCAGGS:E47 | (*Holmberg et al., 2008*) | | |
| Transfected construct (*Mus musculus*) | pCAGGS:E47bm_ ires_GFP | This paper | | See Materials and methods |
| Transfected construct (*Rattus norvegicus*) | pCAGGS:HA-ASCL1 | (*Alvarez-Rodríguez and Pons, 2009*) | | |

*Continued on next page*

*Continued*

| Reagent type (species) or resource | Designation | Source or reference | Identifiers | Additional information |
|---|---|---|---|---|
| Transfected construct (*Rattus norvegicus*) | pCAGGS:HA-NGN1 | This paper | | See Materials and methods |
| Transfected construct (*Mus musculus*) | pCAGGS:ID2_ires_GFP | This paper | | See Materials and methods |
| Transfected construct (*Rattus norvegicus*) | pCAGGS:NGN1 | Other | | Gift from François Guillemot, Francis Crick Institute, London, UK |
| Transfected construct (*Rattus norvegicus*) | pCAGGS:NGN1-t-E47_ires_GFP | This paper | | See Materials and methods |
| Transfected construct (*Rattus norvegicus*) | pCAGGS:NGN1-t-NGN1_ires_GFP | This paper | | See Materials and methods |
| Transfected construct (*Rattus norvegicus*) | pCAGGS:NGN2 | Other | | Gift from François Guillemot, Francis Crick Institute, London, UK |
| Transfected construct (*Rattus norvegicus*) | pCAGGS:SMAD5-SD_ires_GFP | (*Le Dréau et al., 2012*) | | |
| Transfected construct (*Mus musculus*) | pCAGGS:TCF12 | (*Holmberg et al., 2008*) | | |
| Transfected construct (*Mus musculus*) | pCMV2:Flag-E47-RFP | (*Mehmood et al., 2009*) | | |
| Transfected construct (*Mus musculus*) | pCMV2:Flag-E47Δnls-RFP | (*Mehmood et al., 2009*) | | |
| Transfected construct (human) | pCS2:H2B-GFP | (*Le Dréau et al., 2014*) | | |
| Transfected construct (*Mus musculus*) | pCS2:Somitabun | (*Beck et al., 2001*) | | |
| Transfected construct (*Mus musculus*) | pMiW:myc-NGN1 | (*Gowan et al., 2001*) | | |
| Transfected construct (*Mus musculus*) | pMiW:myc-NGN1-AQ | (*Gowan et al., 2001*) | | |
| Transfected construct (empty) | pSuper | Oligoengine | | |
| Transfected construct (empty) | pShin | (*Kojima et al., 2004*) | | |
| Transfected construct (*Gallus gallus domesticus*) | sh-Bmp7 | (*Le Dréau et al., 2012*) | | |
| Transfected construct (*Gallus gallus domesticus*) | sh-Smad1 | (*Le Dréau et al., 2012*) | | |
| Transfected construct (*Gallus gallus domesticus*) | sh-Smad5 | (*Le Dréau et al., 2012*) | | |

*Continued on next page*

*Continued*

| Reagent type (species) or resource | Designation | Source or reference | Identifiers | Additional information |
|---|---|---|---|---|
| Transfected construct (*Gallus gallus domesticus*) | sh-Id2 | This paper | | See Materials and methods |
| Transfected construct (*Mus musculus*) | pTubb3:luc | This paper | | See Materials and methods |
| Transfected construct (*Mus musculus*) | pId2(backbone):luc | (*Kurooka et al., 2012*) | | |
| Transfected construct (*Mus musculus*) | pId2(Full):luc | (*Kurooka et al., 2012*) | | |
| Transfected construct (*Mus musculus*) | pId2(BREonly):luc | (*Kurooka et al., 2012*) | | |
| Transfected construct (*Mus musculus*) | pId2(BREmut):luc | (*Kurooka et al., 2012*) | | |
| Transfected construct (*Gallus gallus domesticus*) | pSox2:luc | (*Saade et al., 2013*) | | |
| Transfected construct (artificial) | pKE7:luc | (*Akazawa et al., 1995*) | | |
| Transfected construct (*Mus musculus*) | pNeuroD:luc | Other | | Gift from François Guillemot, Francis Crick Institute, London, UK |
| Transfected construct (*Mus musculus*) | pNeuroDmut:luc | This paper | | See Materials and methods |
| Transfected construct (*Mus musculus*) | pDll1-M:luc | (*Castro et al., 2006*) | | |
| Transfected construct (*Mus musculus*) | pDll1-N:luc | (*Castro et al., 2006*) | | |
| Transfected construct (*Gallus gallus domesticus*) | pSox2:eGFP | (*Saade et al., 2013*) | | |
| Transfected construct (*Mus musculus*) | pTis21:RFP | (*Saade et al., 2013*) | | |
| Biological sample (*Gallus gallus domesticus*) | developing spinal cord (HH14-HH24) | | | |
| Biological sample (*Gallus gallus domesticus*) | developing telencephalon (E3-E5) | | | |
| Antibody | anti-Ascl1 (mouse) | BD Pharmingen | cat #556604; RRID:AB_396479 | |
| Antibody | Anti-Chx10 (guinea pig) | Other | | Gift from Sam Pfaff, The Salk Institute for Biological Studies, La Jolla, CA, USA; 4% PFA fixation; dilution 1/10,000 |

*Continued on next page*

Continued

| Reagent type (species) or resource | Designation | Source or reference | Identifiers | Additional information |
|---|---|---|---|---|
| Antibody | Anti-E2A (rabbit) | Santa Cruz | cat #sc-349; RRID:AB_675504 | |
| Antibody | Anti-En-1 (mouse) | Other | | Gift from Alexandra Joyner, Sloan Kettering Institute, New York, NY, USA; 4% PFA fixation; dilution 1/100 |
| Antibody | Anti-Evx1 (mouse) | DSHB | cat #99.1-3A2; RRID:AB_528231 | |
| Antibody | Anti-Gata3 (mouse) | Santa Cruz | cat #sc-268; RRID:AB_2108591 | |
| Antibody | Anti-HA (rabbit) | Abcam | cat #ab20084 RRID:AB_445319 | |
| Aantibody | Anti-HuC/D (mouse) | Life Technologies | cat #A-21271; RRID:AB_221448 | |
| Antibody | Anti-ID2 (mouse) | Abcam/Thermo Fisher Scientific | cat #ab166708; RRID: AB_2538566 | |
| Antibody | Anti-ID2 (rabbit) | Calbio Reagents | cat #M213; RRID: AB_1151771 | |
| Antibody | Anti-Isl1 (mouse) | DSHB | cat #402D6; RRID: AB_528315 | |
| Antibody | Anti-Lbx1 (guinea pig) | Other | | Gift from Thomas Müller and Carmen Birchmeier, Max Delbrück Center for Molecular Medicine, Berlin, Germany; 4% PFA fixation; dilution 1/10,000 |
| Antibody | Anti-Lhx1/5 (mouse) | DSHB | cat #4F2; RRID: AB_2314743 | |
| Antibody | Anti-Lhx2/9 (rabbit) | Other | | Gift from Thomas Jessell, Zuckerman Institute, Departments of Neuroscience, and Biochemistry and Molecular Biophysics, Columbia University, New York, NY, USA; 4% PFA fixation; dilution 1/5,000 |
| Antibody | Anti-Lmx1b (mouse) | DSHB | cat #50.5A5; RRID:AB_2136850 | |
| Antibody | Anti-Neurogenin-1 (rabbit) | Millipore | cat #AB15616; RRID:AB_838216 | |
| Antibody | Anti-Nkx6.1 (mouse) | DSHB | cat #F55A10; RRID:AB_532378 | |
| Antibody | Anti-Pax2 (rabbit) | Zymed | cat #71–6000; RRID:AB_2533990 | |
| Antibody | Anti-Pax6 (mouse) | DSHB | Cat#pax6; RRID:AB_528427 | |
| Antibody | Anti-Pax6 (rabbit) | Covance/Biolegend | cat #901301; RRID:AB_2565003 | |

*Continued*

| Reagent type (species) or resource | Designation | Source or reference | Identifiers | Additional information |
|---|---|---|---|---|
| Antibody | Anti-pH3 (rat) | Sigma Aldrich | cat #H9908; RRID:AB_260096 | |
| Antibody | Anti-RFP (rabbit) | (*Herrera et al., 2014*) | | |
| Antibody | Anti-Sox2 (rabbit) | Abcam | cat #ab97959; RRID:AB_2341193 | |
| Antibody | Anti-Trb1 (rabbit) | Abcam | cat #ab31940; RRID:AB_2200219 | |
| Antibody | Anti-Tbr2 (rabbit) | Abcam | cat #ab23345; RRID:AB_778267 | |
| Antibody | Anti-Tlx3 (guinea pig) | Other | | Gift from Thomas Müller and Carmen Birchmeier, Max Delbrück Center for Molecular Medicine, Berlin, Germany; 4% PFA fixation; dilution 1/10,000 |
| Antibody | Anti-Tubulin, beta (mouse) | Millipore | cat #MAB3408; RRID:AB_94650 | |
| Sequence-based reagent | Primers for ChIP of the pKE7 promoter region: | This study | | #FW:CAggTgg CAggCAgg; #REV: AAgCAgAACg Tgggg |
| Sequence-based reagent | Primers for ChIP of the pNeuroD promoter region: | This study | | #FW:CTggCTTCTAA TCTCCATCAC; #REV:gTTTCCAg ATgAggTCgT |
| Sequence-based reagent | Primers for ChIP of the Luciferase ORF: | This study | | #FW:gAAgACgC CAAAAACATA; #REV:CgTATCTCT TCATAgCCTTA |
| Commercial assay or kit | CellTrace Violet Cell Proliferation Kit | Thermo Fisher Scientific | Cat #C34557 | |
| Software, algorithm | Adobe Photoshop CS5 | Adobe | RRID:SCR_014199 | |
| Software, algorithm | FlowJo | TreeStar Inc | RRID:SCR_000410 | |
| Software, algorithm | ImageJ | (*Schneider et al., 2012*) | RRID: SCR_003070 | http://imagej.net/Welcome |
| Software, algorithm | Leica LAS AF Image Acquisition | Leica Microsystems | RRID:SCR_013673 | |

## In ovo electroporation

Fertilized white Leghorn chicken eggs were provided by Granja Gibert, rambla Regueral, S/N, 43850 Cambrils, Spain. Eggs were incubated in a humidified atmosphere at 38·C in a Javier Masalles 240N incubator for the appropriate duration and staged according to the method of Hamburger and Hamilton (HH, (*Hamburger and Hamilton, 1951*). According to EU animal care guidelines, no IACUC approval was necessary to perform the experiments described herein, considering that the embryos used in this study were always harvested at early stages of embryonic development (at E5 at the latest). Sex was not identified at these stages.

Unilateral *in ovo* electroporations in the developing chick spinal cord and dorsal telencephalon were performed respectively at stages HH14-15 and HH18 (54 and 69 hr of incubation). In the telencephalon, corticogenesis was studied specifically in the dorsal-medial-lateral (dML) subregion to

minimize any possible variability along the medial-lateral axis. Plasmids were diluted in RNAse-free water at the required concentration [0 to 4 µg/µl] and injected into the lumen of the caudal neural tube or the right cerebral ventricle using a fine glass needle. Electroporation was triggered by applying 5 pulses of 50 ms at 22.5 V with 50 ms intervals using an Intracel Dual Pulse (TSS10) electroporator. Electroporated chicken embryos were incubated back at 37C and recovered at the times indicated (16–48 hr post-electroporation).

## Cell lines

The human embryonic kidney-derived HEK293T cell line was obtained from ATCC (#CRL-3216; STR profile = CSF1PO: 11,12; D13S317: 12,14; D16S539: 9,13; D5S818: 8,9; D7S820: 11; TH01: 7, 9.3; TPOX: 11; vWA: 16,19; Amelogenin: X). This cell line is not listed in the commonly misidentified cell lines list from the International Cell Line Authentication Committee. Cells were Mycoplasma-free, as routinely assessed using the LookOut Mycoplasma PCR Detection Kit (Sigma #MP0035-1KT). HEK293T cells were grown at sub-confluent density in DMEM supplemented with 10% fetal bovine serum and penicillin/streptomycin at 37°C, 5% CO2.

## Plasmids

To facilitate comparisons in gain-of-function experiments, all the constructs used in this study were inserted under the control of a pCAGGS promoter that harbors high activity in chick (pCAGGS or pCAGGS_ires_GFP, kindly provided by Andy McMahon, *Megason and McMahon, 2002*), and were electroporated at similar concentrations (0, 0.1, 0.5 or 1 µg/µl as specified in the respective figure legends). Non-fluorescent pCAGGS plasmids were combined with 0.25 µg/µl of pCS2_H2B-GFP for visualization. The pCAGGS:ASCL1, pCAGGS:NEUROG1 and pCAGGS:NEUROG2 plasmids were kindly provided by François Guillemot. The pCAGGS:ATOH1_ires_GFP plasmid was obtained by subcloning from a pCMV:ATOH1 kindly provided by Nissim Ben-Arie (*Krizhanovsky et al., 2006*). The pCAGGS:E47 and pCAGGS:TCF12 were kindly provided by Jonas Muhr (*Holmberg et al., 2008*). The pCAGGS:E47bm_ires_GFP plasmid was derived from a pGK:E47_CFP plasmid kindly provided by Yuan Zhang (*Zhuang et al., 1998*). The pCAGGS:ID2_ires_GFP was derived from a pCMV:ID2 plasmid, and the pCAGGS_SMAD5-SD_ires_GFP was described previously (*Le Dréau et al., 2012*). Only E47Δnls-RFP and E47-RFP (in a pFlag-CMV2 vector, kindly provided by Yoshihiro Yoneda, *Mehmood et al., 2009*), Somitabun (pCS2:Somitabun, kindly provided by Jonathan Slack, *Beck et al., 2001*) and NEUROG1-AQ and its wild-type NEUROG1 version (pMiW:myc-NGN1 and pMiW:myc-NGN1-AQ, kindly gifted by Jane Johnson, *Gowan et al., 2001*) were used in a different backbone. HA-tagged versions of ASCL1 (pCAGGS:HA-ASCL1, *Alvarez-Rodríguez and Pons, 2009*) and NEUROG1 (pCAGGS:HA-NEUROG1) and a Flag-E47-RFP construct were used for chromatin immuneprecipitation assays. Inhibition of *cBmp7, cSmad1, cSmad5 or cId2* expression was triggered by electroporation of short-hairpin constructs inserted into the pSuper (Oligoengine) or pSHIN (*Kojima et al., 2004*) vectors. Electroporation of 2–4 µg/µl of these constructs caused a specific and reproducible 50% inhibition of the target expression (see *Le Dréau et al., 2012*). The pSox2:GFP and pTis21:RFP reporters used to assess the modes of divisions undergone by spinal progenitors were previously described in details (*Saade et al., 2013*). The pSox2:luc derived from the pSox2:GFP (*Saade et al., 2013*). The different versions of the pId2:luc reporter were kindly provided by Yoshifumi Yokota (*Kurooka et al., 2012*) and the pkE7:luc by Masashi Kawaichi (*Akazawa et al., 1995*). The pNeuroD:luc, pDll1-M:luc and pDll1-N:luc reporters were kindly provided by François Guillemot (*Castro et al., 2006*). The pNeuroDmut:luc reporter was obtained by site-directed mutagenesis of the single CAGGTG E-box contained in the NeuroD promoter region. The pTubb3:luc reporter was obtained by subcloning the Tubb3 enhancer region present in the pTubb3enh:GFP plasmid (kindly provided by Jonas Muhr, *Bergsland et al., 2011*) into the pGL3:luc vector (Promega). Please head for the Key Resources Table for additional information.

## Generation of tethered constructs

The tethered bHLH dimers were derived from the pCAGGS:ASCL1-t-E47_ires_GFP kindly provided by François Guillemot (*Geoffroy et al., 2009*). This plasmid and pCAGGS:NEUROG1 were used as templates to generate the ASCL1-t-ASCL1, NEUROG1-t-E47 and NEUROG1-t-NEUROG1 constructs inserted into pCAGGS_ires_GFP, using a tether peptide AAAGTSAGGAAAGTSASAATGA flanked

by SpeI and ClaI restriction sites as described previously (*Henke et al., 2009*). Expression of the tethered bHLH dimers was assessed by western blot after transfection into HEK293 cells. Transient cell transfections were obtained by electroporation applying 2 pulses of 120V, 30 ms (Microporator MP-100, Digital Bio). Cells were grown for 24 hr onto poly-L-Lysine-coated 6-well dishes in DMEM/F12 supplemented with 10% fetal bovine serum and 50 mg/L of Gentamicin until reaching 70-80% confluence. The typical transfection efficiency of this procedure was 40–60%. Cells were lysed in 1X SDS loading buffer (10% glycerol, 2% SDS, 100 mM dithiothreitol, and 62.5 mM Tris-HCl, pH 6.8) and DNA was disrupted by sonication. Protein extracts were separated by SDS-PAGE electrophoresis, transferred to Immobilon-FL PVDF membranes (IPFL00010, Millipore), blocked with the Odyssey Blocking Buffer (927–40000, LI-COR), and incubated with antibodies against ASCL1 (BD Pharmingen, cat#556604, 1:1000), NEUROG1 (Millipore, cat#AB15616, 1:3000) or E2A (Santa Cruz, cat#sc-763, 1:1000). Detection was performed using fluorescence-conjugated secondary antibodies and an Odyssey Imaging System (LI-COR).

## Immunohistochemistry

For immunohistochemistry experiments, chicken embryos were carefully dissected, fixed for 2 hr at 4°C in 4% paraformaldehyde and rinsed in PBS. Immunostaining was performed on either vibratome (40 µm) or cryostat (16 µm) sections following standard procedures. After washing in PBS-0.1% Triton, the sections were incubated overnight at 4C with the appropriate primary antibodies (see Key Resources Table) diluted in a solution of PBS-0.1% Triton supplemented with 10% bovine serum albumin or sheep serum. After washing in PBS-0.1% Triton, sections were incubated for 2 hr at room temperature with the appropriate secondary antibodies diluted in a solution of PBS-0.1% Triton supplemented with 10% bovine serum albumin or sheep serum. Alexa488-, Alexa555- and Cy5-conjugated secondary antibodies were obtained from Invitrogen and Jackson Laboratories. Sections were finally stained with 1 µg/ml DAPI and mounted in Mowiol (Sigma-Aldrich).

## Image acquisition, treatment and quantification

Optical sections of fixed samples (transverse views of the spinal cord, coronal views for the telencephalon) were acquired at room temperature with the Leica LAS software, in a Leica SP5 confocal microscope using 10x (dry HC PL APO, NA 0.40), 20x (dry HC PL APO, NA 0.70), 40x (oil HCX PL APO, NA 1.25–0.75) or 63x (oil HCX PL APO, NA 1.40–0.60) objective lenses. Maximal projections obtained from 2 µm Z-stack images were processed in Photoshop CS5 (Adobe) or ImageJ for image merging, resizing and cell counting.

Quantification of endogenous ID2 intensity was assessed using the ImageJ software. Cell nuclei of H2B-GFP + electroporated and neighboring non-electroporated cells were delimited by polygonal selection, and the mean intensity of ID2 immunoreactivity quantified as mean gray values. Quantifications were performed on at least six electroporated and six non-electroporated cells per image, in at least three different images per embryo.

## In situ hybridization

Chicken embryos were recovered at the indicated stage, fixed overnight at 4°C in 4% PFA, rinsed in PBS and processed for whole mount RNA in situ hybridization following standard procedures. Probes against chick cId2 (#chEST852M19) and cNeurog2 (#chEST387d10) were purchased from the chicken EST project (UK-HGMP RC). Probes against cTcf3/E2a, cAscl1 and cNeurog1 were kindly provided by Drs Jonas Muhr, José-Maria Frade and Cristina Pujades. The probe against cTcf12/cHeb was obtained by PCR from genomic DNA of E4 chicken embryonic tissue and the purified 623 nucleotides insert was sub-cloned into the pGEM-T vector (Promega). Hybridized embryos were post-fixed in 4% PFA and washed in PBT. 45µM-thick sections were cut with a vibratome (VT1000S, Leica), mounted and photographed using a microscope (DC300, Leica). The data show representative images obtained from three embryos for each probe.

## Luciferase assay

Transcriptional activity was assessed following electroporation of a luciferase reporter together with a renilla luciferase reporter used for normalization, in combination with the indicated plasmids required for experimental manipulation. Embryos were harvested 24 hr later and GFP-positive neural

tubes were dissected and homogenized in a Passive Lysis Buffer on ice. Firefly- and renilla-luciferase activities were measured by the Dual Luciferase Reporter Assay System (Promega).

## Cell cycle exit assay

The average number of divisions undergone by electroporated spinal progenitors was assessed in vivo using the CellTrace Violet Cell Proliferation Kit (Invitrogen). The Violet cell tracer (1 mM), a cytoplasmic retention dye that becomes diluted as cells divide, was injected into the lumen of the neural tube at the time of electroporation. Embryos were recovered 48 hr later, the neural tubes were carefully dissected and recovered and the cells dissociated following a 10–15 min digestion in Trypsin-EDTA (Sigma). The fluorescence intensity of the Violet tracer was measured in viable dissociated electroporated GFP$^+$ cells in the 405/450 nm excitation/emission range on a Gallios flow cytometer (Beckman Coulter, Inc).

## Assessment of the modes of divisions

Chicken embryos were recovered 16 hr after co-electroporation of the pSox2:eGFP and pTis21:RFP reporters together with the indicated bHLH TF-encoding plasmids. Cell suspensions were obtained from pools of 6–8 dissected neural tubes after digestion with Trypsin-EDTA (Sigma) for 10–15 min, and further processed on a FACS Aria III cell sorter (BD Biosciences) for measurement of eGFP and RFP fluorescences. At least 1,000 cells for each progenitor population (PP, PN and NN) were analyzed per sample.

## Chromatin immunoprecipitation assay

HEK293T cells were transfected by a standard calcium phosphate co-precipitation protocol with combinations of pCAGGS_ires_GFP, pCAGGS:HA-ASCL1, pCAGGS:HA-NEUROG1, pCMV2_Flag-E47-RFP together with the pkE7:luc or pNeuroD:luc reporters, with a total of 10 µg of DNA per 100 mm dish. 24 hr later, cells were collected and 10% of the material was reserved to check transfection by Western blot. For chromatin immunoprecipitation assays, approximately 1 million transfected HEK293T cells were fixed with 1% formaldehyde for 10 min at room temperature. Fixation was quenched by adding 0.125M glycine for 5 min. After two washes with PBS, cells were lysed on ice for 20 min in a lysis buffer containing protease inhibitors (1% SDS; 10 mM EDTA pH8.0; 50 mM Tris-HCl pH8.1). Sonication was performed with a Bioruptor sonicator to obtain 200–500 bp shredded chromatin fragments. Chromatin purification was carried out by spinning samples down at maximum speed at 4C during 30 min. Purified chromatin was pre-cleared with protein A agarose (Millipore #16–125) for 30 min. 25 µg of chromatin were immunoprecipitated with 5 µL of anti-RFP serum (*Herrera et al., 2014*), 2 µg of anti-HA (Abcam, cat#20084), anti-NEUROG1 (Millipore, cat#15616) or unspecific rabbit IgG (Diagenode, cat#C15410206) antibodies. Antibody-chromatin complexes were recovered using magnetic beads (Magna ChIP, Millipore, cat#16–661) and immuno-complexes were washed once with TSE I (0.1% SDS; 1% Triton-X100; 2 mM EDTA pH8.0; 20 mM Tris-HCl pH8.1; 150 mM NaCl), TSE II (0.1% SDS; 1% Triton-X100; 2 mM EDTA pH8.0; 20 mM Tris-HCl pH8.1; 500 mM NaCl), TSE III (0.25M LiCl; 1% NP-40; 1% Sodium Deoxicholate; 1 mM EDTA pH8.0; 10 mM Tris-HCl pH8.1) and twice with TE (Tris-HCl 10 mM, EDTA 1 mM). Reversal of crosslinking was done by incubating samples in elution buffer (1% SDS, 0.1M NaHCO3) overnight at 65C. DNA was purified by phenol-chloroform extraction followed by ethanol precipitation. Quantification of the DNA target regions and negative control (luciferase ORF) was assessed by qPCR in a Lightcycler 480 (Roche) using specific primers (see Key Resources Table).

Proteins extracts were obtained by incubation in a RIPA buffer (150 mM NaCl, 1.0% NP-40,0.5% sodium deoxycholate, 0.1% SDS and 50 mM Tris, pH 8.0) supplemented with protease and phosphatase inhibitors for 20 min on ice and centrifugation (20 min at maximum speed). 30 µg of protein samples were mixed with the Laemmli buffer (375 mM Tris pH = 6.8, 12%SDS, 60% glycerol, 600 mM DTT, 0.06% bromphenol blue), heated to 95°C and then separated on a SDS-PAGE gel in running buffer (25 mM Tris base, 190 mM glycine, 0.1% SDS, pH = 8,3). Proteins were transferred to a nitrocellulose membrane using transfer buffer (190 mM glycine, 25 mM Tris, 20% Methanol, 0.1% SDS) for 90 min at 80V. Membranes were blocked for 1 hr with a solution of PBS-5% milk, 1% Tween (PBST) and further incubated overnight at 4C with appropriate primary antibodies diluted in PBST: rabbit anti-HA (Abcam, cat #ab20084), rabbit anti-RFP serum (Herrera et al, 2014) and mouse anti-

Tubulin beta (Millipore, cat #MAB3408). After three washes in PBST, membranes were incubated with Horseradish peroxidase-conjugated anti-rabbit IgG or anti-mouse IgG secondary antibodies (Sigma-Aldrich, cat#GENA934-1ML and cat#GENA931) for 1 hr at room temperature and the signals detected by chemiluminescence using the Immobilon western chemiluminiscent HRP substrate (Sigma-Aldrich, cat# WBKLS0100).

## Statistical analyses

No statistical method was used to predetermine sample size. The experiments were not randomized. The investigators were not blinded to allocation during experiments or outcome assessment. Statistical analyses were performed using the GraphPad Prism six software (GraphPad Software, Inc.). For in vivo experiments, cell counts were typically performed on 2–5 images per embryo and $n$ values correspond to different embryos, except for the assessment of the modes of divisions where $n$ values correspond to pools of embryos. For in vitro chromatin immunoprecipitation assays, $n$ values represent the numbers of independent experiments performed. The $n$ values are indicated in the corresponding figure legends. The normal distribution of the values was assessed by the Shapiro-Wilk normality test. Significance was then assessed with a two-sided unpaired t-test, one-way ANOVA + Tukey's test or two-way ANOVA + Sidak's test for data presenting a normal distribution, or alternatively with non-parametric Mann–Whitney or Kruskal-Wallis +Dunn's multiple comparisons' tests for non-normally distributed data. n.s: non-significant; *: $p < 0.05$ or less, as indicated in individual figures.

## Acknowledgements

We thank the members of the laboratory for helpful comments on the study. We are grateful to Drs N Ben-Arie, C Birchmeier, J-M Frade, F Giraldez, F Guillemot, TM Jessell, JE Johnson, A Joyner, M Kawaichi, A McMahon, J Muhr, T Müller, S Pfaff, C Pujades, J Slack, Y Yokota, Y Yoneda and Y Zhuang for kindly providing reagents. We also thank the Developmental Studies Hybridoma Bank, developed under the auspices of the NICHD and maintained by The University of Iowa, Department of Biological Sciences, Iowa City, IA, USA. We acknowledge E Rebollo and the IBMB Molecular Imaging platform and J Comas and the PCB Flow Cytometry facility for excellent assistance. This work was supported by the grants to EM from BFU2016-81887-REDT and BFU2016-77498-P.

## Additional information

### Funding

| Funder | Grant reference number | Author |
|---|---|---|
| Asociación Española Contra el Cáncer | AIO2014 | Gwenvael Le Dréau |
| Consejo Nacional de Ciencia y Tecnología | | René Escalona |
| Ministerio de Educación, Cultura y Deporte | #FPU13/01384 | Raquel Fueyo |
| Ministerio de Economía y Competitividad | #FJCI-2015-26175 | Antonio Herrera |
| Ministerio de Economía y Competitividad | BFU2014-53633-P | Sebastian Pons |
| Ministerio de Economía y Competitividad | BFU2015-69248-P | Marian A Martinez-Balbas |
| Fondation Jérôme Lejeune | Fondation Jérôme Lejeune.2016 | Marian A Martinez-Balbas |
| Ministerio de Economía y Competitividad | BFU2016-81887-REDT | Elisa Marti |
| Ministerio de Economía y Competitividad | BFU2016-77498-P | Elisa Marti |

The funders had no role in study design, data collection and interpretation, or the decision to submit the work for publication.

## Author contributions
Gwenvael Le Dréau, Conceptualization, Formal analysis, Supervision, Investigation, Visualization, Methodology, Writing—original draft; René Escalona, Formal analysis, Investigation; Raquel Fueyo, Antonio Herrera, Formal analysis, Investigation, Methodology; Juan D Martínez, Susana Usieto, Anghara Menendez, Methodology; Sebastian Pons, Marian A Martinez-Balbas, Resources, Supervision; Elisa Marti, Resources, Supervision, Funding acquisition, Project administration, Writing—review and editing

## Author ORCIDs
Gwenvael Le Dréau (iD) http://orcid.org/0000-0002-6877-3670
Raquel Fueyo (iD) http://orcid.org/0000-0001-7106-7163
Elisa Marti (iD) https://orcid.org/0000-0001-5839-7133

## Decision letter and Author response
Decision letter https://doi.org/10.7554/eLife.37267.023
Author response https://doi.org/10.7554/eLife.37267.024

# Additional files

## Supplementary files
• Transparent reporting form
DOI: https://doi.org/10.7554/eLife.37267.021

## Data availability
All data generated or analysed during this study are included in the manuscript and supporting files.

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
