## [Decision Letter]

[Editors’ note: this article was originally rejected after discussions between the reviewers, but the authors were invited to resubmit after an appeal against the decision.]

Thank you for submitting your manuscript "E proteins differentially co-operate with proneural bHLH transcription factors to sharpen neurogenesis" to *eLife*. Your manuscript has been reviewed by a Senior Editor, a Reviewing Editor, and three reviewers.

While the reviewers found the work interesting, they also raised a large number of substantive questions that left us no choice but to reject the paper. This decision reflects the *eLife* view that a submitted manuscript should be reasonably close to a final version to be accepted. We hope that the reviewers' comments will be useful to you. We apologize for not being able to deliver better news, and we hope that you will continue to consider *eLife* for future submissions.

Reviewer #1:

In the manuscript by Le Dreau, Escalona and colleagues, the authors have assessed the function and mechanisms of action of E proteins and bHLH transcription factors during neurogenesis and neural fate specification. Using predominantly chick neural tube electroporation assays, as well as molecular approaches, the authors present a model in which E proteins have differential effects on neurogenesis, depending whether they interact with ASCL1/ATOH1 versus NEUROG1/2 bHLH transcription factors. In addition, the authors argue that Bmp signaling contributes to this specificity through induction of ID2 expression, which inhibits the activity of ASCL1/ATOH1 bHLHs.

The authors have performed an impressive number of experiments to generate a model of how various HLH proteins interact to generate the appropriate number of neurons of a specific fate. However, I found the amount of data somewhat overwhelming, and in places disjointed with the overall point the authors are trying to make. In many places, I do not think the authors went far enough to support their model, or provide enough data to convince me their results have biological meaning.

In Figure 1 the authors show that inhibiting Bmp5 or Smad signaling leads to a selective loss of p2-domain interneurons, as assessed by Chx10 and Gata3 expression. These neurons are defined by Ascl1 expression in progenitors. First, it is unclear to me why these cells would be sensitive to Bmp loss, as most Bmps are normally expressed in the dorsal region of the spinal cord. Second, what is the evidence that p2-domain cells are normally responding to Bmp? Is there detectable expression of phospo-Smad1/5 in this domain? Finally, what happens to expression of Ascl1 under these conditions? Most of the markers analyzed are expressed in postmitotic cells, and it is unclear what is happening to the identity of spinal progenitors in these experiments.

Figure 2 addresses the role of Id2 proteins, where the authors first state the Bmp is necessary and sufficient to promote Id2 expression in the spinal cord. This is not actually shown for the endogenous gene but instead using an Id2 promoter-luciferase reporter that was electroporated within the spinal cord. So, I think this is somewhat overstated in the text. Based on the data in Figure 1, one also wonders what happens to endogenous Id2 expression after Bmp7 and Smad1/5 knockdown.

The main part of this Figure analyzes the effects of Id2 misexpression and inhibition, where the authors find that Id2 misexpression inhibits the generation of most ventral cell types, while knockdown of Id2 is argued to cause premature differentiation. The biological relevance of these experiments is unclear to me, largely because there is clearly a domain in the ventral spinal cord that is lacking in Id2 expression (Figure 2B-D). If this domain includes the p2/Ascl1 domain then it hard to understand how these experiments provide insights into what is happening naturally.

In the next experiments the authors asked whether Id2 contribute to neurogenesis through sequestering E proteins. The authors show that overexpression of E proteins increases the production of differentiated cells, an activity that can be inhibited by co-expression of Id2. They also expressed a dominant negative E protein that inhibited neural differentiation, and this effect was lost when Id2 was depleted.

In Figure 4, the authors argue that E47 has selective function in the generation of neurons derived from ASCL1+ domains. This is partly based on experiments expressing dnE47 that the author's state has a more selective effect on ASCL1 derivatives. The quantification of these experiments is based on ratios of total neurons from electroporated and non-electroporated sides of the spinal cord. These results could conceivably reflect the dorsal-ventral gradients of electroporation efficiencies that one normal sees. Ideally one would want to know how these data look in terms of electroporated (GFP+) neurons that expressed a specific marker. If for example, if motor neurons are unaffected, one would expect to see a larger percentage of Isl1+/GFP+ than Chx10+/V2+ neurons in these experiments.

In Figure 5 the authors provide evidence that the specificity of ASCL1/E47 action are due to its ability to selectively activate sequences containing CAGSTG sequences. E47 also inhibited the ability of NEUROG1 activate its CADATG targets, suggested that E47 can functionally inhibit NEUROG1 at certain sequences.

Finally, the authors tested whether differentiation interactions of ASCL1 and NEUROG1 with E47 might operate in other regions of the CNS. The authors show that E47 enhances the ability of ASCL1 promote neuronal differentiation in the cortex. In contrast they found that E47 blocks the ability of NEUROG1 to promote cortical neuronal differentiation. As with many other experiments in this study, the biological relevance of these interactions are unclear, as we are given no detailed information on the temporal or spatial expression profile of E47 during cortical development.

The authors have provided detailed information on the statistical methods used in the methods section, and n values for samples in the Figure legends.

Reviewer #2:

This is a interesting study that puts forward a new mechanistic model to understand how different/limiting levels of E-proteins could have differential effects on neurogenesis through modulating activity of proneural bHLH proteins. The model is supported by manipulating levels of BMP signaling, ID levels, and E-protein/proneural bHLH ratios in the chick neural tube and assessing consequences to division patterns and neuronal subtype specification during embryogenesis. Additional experiments assess the different activities on the proneural bHLH factors with and without the E-protein E47 with respect to transcriptional activity of reporter genes with different motifs. The study is written clearly, and data are shown carefully and thoroughly for the most part. The ChIP data are not particularly strong, and the activities rely heavily on over expression assays, nevertheless, multiple types of experiments are used to develop the model and support the conclusions presented.

1) The authors come up with the phrase 'dual co-operation of E proteins with proneural proteins' to describe the differential consequences to neurogenesis that result when E-proteins are limiting. It is a confusing description. Based on the model put forward, it is a differential activity of homodimers of ASCL1/ATOH1 versus NEUROG1/NEUROG2 that form when E-proteins are limiting reflecting the intrinsic binding preferences of the different proneural bHLH TFs. To call this differential co-operation of E-proteins does not really help understand what is going on. I recommend changing how this is stated throughout the manuscript.

2) Some of the differences in activities could be affected by different levels of the proneural proteins made from the expression constructs. Assessment of the protein levels will be difficult to compare because no tag is included, and the antibodies can work at different levels. Minimally it should be considered. Ideally it should be evaluated. This only becomes important because arguments are made about the different effects of E-proteins on the factors. It may be possible that the protein levels of the proneurals are substantially different and are showing differential effects in these assays. This could be true even though pCIG is used for them all.

Also related to this issue, in the Discussion section it is stated that: "Such co-operation appears to be particularly crucial in the case of ASCL1, whose overexpression could barely increase neurogenic divisions per se." This contrasts with other studies ie Nakada 2014 that show Ascl1 is very efficient at increasing differentiation even at 24 hrs. Is it possible that expression levels are relatively low from the construct used here relative to Nakada, or relative to the other bHLH in the current study as mentioned above? This may need some consideration or ruled out by determining the comparative levels of the overexpressed bHLH proteins.

3) All the results in the study are from E47 but the conclusions are generalized to E-proteins. Was HEB tested in these assays? This should be shown if the generalization is being used.

*Reviewer #3:*

In this study, LeDréau et al., explore how differences in the formation of proneural bHLH-E protein transcription factor complexes can influence neuronal differentiation in the developing spinal cord and cortex. A link to BMP influences on this process is further explored though an examination of the function of Id2, a BMP-regulated inhibitor of proneural bHLH-E protein complexes. Some of the initial observations include findings that shRNA knockdown of Bmp7, Smad1, or Smad5 potently suppresses the formation of certain classes of spinal neurons (dI1, dI3, and V2a/b) while having less of an effect on other classes (dI6, V0, V1, and MNs). These trends are argued to align with the association of these two groups with the activity of the proneural factors Ascl1 and Atoh1 vs. Neurog1 and Neurog2. The connection of BMP signaling to proneural genes is supposed by evidence that BMP7/Smad1/5 can regulate Id2. shRNA knockdown of Id2 appears to broadly enhance neurogenic differentiation.

The focus of the study then shifts to an exploration of the requirement and sufficiency of E proteins, primarily E47, for differentiation of different classes of neurons. The key finding is an unexpected discrepancy in how E47 modulates the neurogenic activity of Ascl1/Atoh1 vs. Neurog1/Neurog2. E47 potentiates the activities of Ascl1/Atoh1 while showing an inhibitory effect on Neurog1/Neurog2 in some assays. It is further argued that Neurog1/Neurog2 but not Ascl1/Atoh1 may primarily function as homodimeric complexes and that E47 addition may shift the capacity of the bHLH proteins to bind to certain E box motifs, favoring interactions with CAGSTG sites vs. CADATG sites. Lastly, the authors show that the differential cooperativity of Ascl1 and E47 vs. Neurog1/2 and E47 also holds true in the embryonic cortex, suggesting that this may be a widespread phenomenon.

Overall, the study makes some interesting and important observations. First, E proteins do not universally increase the activity of proneural bHLH proteins, a presumption that comes from classic studies of the cooperativity of myogenic bHLH factor and E proteins. Second, the study argues that certain proneural bHLH proteins such as Neurog1 and Neurog2 may function as homodimeric complexes while others are heterodimeric, which has implications for whether the complex could be regulated by Id proteins. However, the study also has significant weaknesses, most notably inconsistencies with some results and the conclusions/models that are drawn. In addition, while it is suggested that the DNA binding specificity of Neurog1/2 may be altered by the addition of E47, conclusive proof is not provided. Lastly, the studies showing the importance of E47 for neurogenesis are based on misexpression studies without complementary loss of function data. It thus remains unclear whether Neurog1/Neurog2 functions are indeed independent of E protein activities as proposed.

1) A central premise of the study is that BMP signaling differentially regulates neurogenesis by controlling Id2 expression. However, the Id2 shRNA experiments in Figure 2 seems to show broader effects of Id2 loss encompassing both Ascl1/Atoh1 and Neurog1/Neurog2-associated groups. How does one reconcile this discrepancy with the overall model?

2) The inhibitory effects of E47 expression appear to be only transient- in Figures4K and 4M, suppression is seen at the 24 h time point, but not at 48 h. How does this difference play into the central model?

3) Evidence that E47 inhibits Neurog1/2 function solely comes from gain of function experiments, which can be artifactual. The authors should examine the consequences of E47 loss using shRNA or CRISPR mutation approaches. If it is true that Neurog1/2 act as homodimers, then there should be little change in the differentiation of the Neurog1/2-associated neuron groups, while Ascl1/Atoh1-associated groups should be severely impacted.

4) The stoichiometry of various proneural proteins and E protein needs to be carefully considered- too much of either the proneural factor or E protein could throw off activities. Ideally, the amount used should be titrated to identify the optimal amounts to be used. It might be further useful to judge whether equivalent amounts of each protein are present in the experiments using epitope tagged versions of the proteins and immunoblotting (or semi-quantitative fluorescence microscopy).

5) The authors show that another E protein, Tcf12 is also expressed in the spinal cord, yet few experiments are performed to explore its relevance. Does Tcf12 similarly provide enhancement of Ascl1/Atoh1 function while inhibiting Neurog1/2?

6) The data collectively suggest that the presence of E47 alters the preferences of proneural bHLH proteins for certain E box motifs, yet direct proof remains lacking. These conclusions could be strengthened by the inclusion of DNA binding experiments (EMSA) showing that the mixing of proneural proteins with E47 changes their affinity for certain E boxes as is implicated in the Figure 7 model. Id proteins could also be added in to test whether certain complexes (such as Neurog1/2 homodimers) are indeed resistant to Ids. Another way to achieve these results might be to build out the use of reporter constructs adding modified versions of the pkE7 construct containing various E box motifs and ask whether there are switches in the activity levels of the different reporters.

7) The ChIP studies shown in Figure 5 are performed in 293 cells, while all of the luciferase assays and cell differentiation experiments are conducted in the chick spinal cord or brain. It would be best to perform ChIP in a system where some signs that the proneural bHLH factors with or within E47 are transcriptionally active.

8) The studies here do not address the secondary wave of proneural proteins, i.e. Neurod1 and Neurod4 which come on as cells begin to differentiate. Is the expression and/or function of these factors also impacted by E47 presence/absence and Id2 inhibition?

---

## [Author Response]

[Editors’ note: the author responses to the first round of peer review follow.]

Reviewer #1:In the manuscript by Le Dreau, Escalona and colleagues, the authors have assessed the function and mechanisms of action of E proteins and bHLH transcription factors during neurogenesis and neural fate specification. Using predominantly chick neural tube electroporation assays, as well as molecular approaches, the authors present a model in which E proteins have differential effects on neurogenesis, depending whether they interact with ASCL1/ATOH1 versus NEUROG1/2 bHLH transcription factors. In addition, the authors argue that Bmp signaling contributes to this specificity through induction of ID2 expression, which inhibits the activity of ASCL1/ATOH1 bHLHs.The authors have performed an impressive number of experiments to generate a model of how various HLH proteins interact to generate the appropriate number of neurons of a specific fate. However, I found the amount of data somewhat overwhelming, and in places disjointed with the overall point the authors are trying to make. In many places, I do not think the authors went far enough to support their model or provide enough data to convince me their results have biological meaning.R1C1) In Figure 1 the authors show that inhibiting Bmp5 or Smad signaling leads to a selective loss of p2-domain interneurons, as assessed by Chx10 and Gata3 expression. These neurons are defined by Ascl1 expression in progenitors.

We wish to precise that inhibiting BMP7 or SMAD1/5 activity does not only cause a reduction of p2-derived V2a/V2b interneurons in the ventral part of the developing spinal cord, but also markedly reduces the generation of dorsal dI1, dI3 and dI5 interneurons (previously published in Le Dreau et al., 2012 and presented again in Figure-1B).

R1C2) First, it is unclear to me why these cells would be sensitive to Bmp loss, as most Bmps are normally expressed in the dorsal region of the spinal cord.

While it is true that the expression of BMP ligands and their canonical signaling (dependent on SMAD1/5/8 activity) are confined to the most dorsal part of the developing spinal cord at early stages during neural patterning, the expression of several BMP ligands, such as BMP7, spreads ventrally in the ventricular zone at the onset of interneuron neurogenesis around HH18 in chick embryos (see Annex 1C taken from Le Dreau et al., 2012). Concomitantly, the canonical BMP activity, as assessed by *in ovo* electroporation of a SMAD1/5/8-responsive GFP reporter, can be detected throughout the Dorsal-Ventral (D-V) axis of the developing spinal cord (see Annex 1D and Annex 2G taken from Le Dreau et al., 2014). We believe that these previously published results are strong evidences that the canonical BMP pathway is active throughout the D-V axis of the developing spinal cord from the onset of interneuron generation.

R1C3) Second, what is the evidence that p2-domain cells are normally responding to Bmp? Is there detectable expression of phospo-Smad1/5 in this domain?

In addition to the arguments mentioned above, we had moreover reported that phospho-SMAD1/5/8 staining (which specifically detects the active form of SMAD1/5/8) can be observed in mitotic spinal progenitors along the entire D-V axis from the onset of interneuron generation (see Annex 3A-G taken from Le Dreau et al., 2014). In addition, work from Misra et al., reported that the generation of V2a and V2b interneurons from p2 progenitors is dependent on BMP signaling (Misra et al., 2014, reference included in our manuscript, subsection “The canonical BMP pathway differentially regulates the generation of spinal neurons derived from progenitors expressing ASCL1/ATOH1 or NEUROG1/NEUROG2/PTF1a”). They showed for instance that *in ovo* electroporation of the extracellular BMP antagonist Noggin markedly reduces the generation of both Chx10+ and Gata3+ neurons. Altogether, we believe that these previously published results demonstrate that p2 are submitted to BMP signaling during neurogenesis and depend on it for proper V2a/b generation.

R1C4) Finally, what happens to expression of Ascl1 under these conditions? Most of the markers analyzed are expressed in postmitotic cells, and it is unclear what is happening to the identity of spinal progenitors in these experiments.

In a previous work, we demonstrated that the activity of the canonical BMP pathway spreads ventrally at the onset of interneuron generation (Le Dreau et al., 2012). We further brought evidences that this canonical BMP activity is not required for maintenance of neural patterning as first anticipated (see Annex 4). When we analyzed the expression pattern of the proneural bHLH genes in response to modulations of canonical BMP signaling (using sh-Smad1/5 and constitutively active SMAD1/5 mutants), we did not observe any significant alteration of their D-V pattern (see Annex 5). Later on, we demonstrated that the primary function of the canonical BMP pathway during spinal neurogenesis is to dictate the mode of divisions of spinal progenitors: high levels of SMAD1/5 activity promote symmetric proliferative PP divisions, whereas lower levels enable spinal progenitors to undergo neurogenic divisions (Le Dreau et al., 2014).

Altogether, we believe that the issues raised by the reviewer #1 in his/her first comments were largely addressed and answered in previous studies.

R1C5) Figure 2 addresses the role of Id2 proteins, where the authors first state the Bmp is necessary and sufficient to promote Id2 expression in the spinal cord. This is not actually shown for the endogenous gene but instead using a Id2 promoter-luciferase reporter that was electroporated within the spinal cord. So, I think this is somewhat overstated in the text. Based on the data in Figure 1, one also wonders what happens to endogenous Id2 expression after Bmp7 and Smad1/5 knockdown.

In our previous work, we reported the effects of both SMAD1/5 gain-of-function and inhibition of the canonical BMP activity (mediated by *in ovo* electroporation of Noggin) on the endogenous expression of the four ID members, which showed that expression of *cId1, cId2 and cId3* is dependent on and positively modulated by the canonical BMP pathway (see Annex 6 taken from Le Dreau et al., 2014). We moreover showed that *cId2* transcript levels in PP, PN and NN progenitors follow a gradient that correlates well with the gradient of endogenous SMAD1/5 activity (see panels H-J from Annex 3). These results already argued for *cId2* being directly regulated by the canonical BMP pathway during spinal neurogenesis. While we agree with the reviewer #1 that the additional experiments performed in our present study with the pId2:luc reporter do not demonstrate *per se* that *cId2* is a direct target of the canonical BMP pathway, they however reinforce this idea. These results taken altogether support the notion that ID2 might participate in mediating the function of the canonical BMP pathway during spinal neurogenesis, which we believe was the essential demonstration required to test the involvement of ID2 in this process.

R1C6) The main part of this Figure analyzes the effects of Id2 misexpression and inhibition, where the authors find that Id2 misexpression inhibits the generation of most ventral cell types, while knockdown of Id2 is argued to cause premature differentiation. The biological relevance of these experiments is unclear to me, largely because there is clearly a domain in the ventral spinal cord that is lacking in Id2 expression (Figure 2B-D). If this domain includes the p2/Ascl1 domain then it hard to understand how these experiments provide insights into what is happening naturally.

As stated by the reviewer #1, knockdown of *cId2* causes an overall premature neuronal differentiation. Conversely, ID2 gain-of-function causes an overall delay/reduction in neuronal differentiation (rather than specifically on ventral cell types). We believe that these results are relevant and fit well the classical role assigned to ID factors in inhibiting differentiation.

Concerning the expression pattern of *cId2*, we agree with the reviewer #1 that *cId2* transcripts were not detected in the ventral part of the developing spinal cord at HH14 (Figure 2A), a stage at which neuronal production has not started yet. We added this panel to illustrate that *cId2* expression is restricted to dorsal domains at early developmental stages (when neural patterning is being established) as is the canonical BMP activity (Le Dreau et al., 2012; Tozer et al., 2013). Around the onset of neurogenesis, *cId2* transcripts appear expressed also in ventral areas, in correlation with the ventral spreading of cBmp7 expression and the canonical BMP activity (as explained above in the answer to the points R1C2-4). This ventrally-spread expression pattern was illustrated with the images corresponding to Figure 2C-D.

To answer more accurately the reviewer #1’s concern, we performed additional experiments and now bring new data that show unequivocally that *cId2* transcripts and protein are expressed by all ventral progenitor domains during spinal neurogenesis, except the pMN domain (new Figure—figure supplement 2). The manuscript has been modified accordingly (subsection “ID2 acts downstream of the canonical BMP pathway to differentially regulate the generation of spinal neurons derived from progenitors expressing ASCL1/ATOH1 or NEUROG1/NEUROG2/PTF1a”). We hope these precisions will satisfy the reviewer #1’s concern.

R1C7) In the next experiments the authors asked whether Id2 contribute to neurogenesis through sequestering E proteins. The authors show that overexpression of E proteins increases the production of differentiated cells, an activity that can be inhibited by co-expression of Id2. They also expressed a dominant negative E protein that inhibited neural differentiation, and this effect was lost when Id2 was depleted.

We understand from this comment that the reviewer #1 was convinced by this part of the study.

R1C8) In Figure 4, the authors argue that E47 has selective function in the generation of neurons derived from ASCL1+ domains. This is partly based on experiments expressing dnE47 that the author's state has a more selective effect on ASCL1 derivatives. The quantification of these experiments is based on ratios of total neurons from electroporated and non-electroporated sides of the spinal cord. These results could conceivably reflect the dorsal-ventral gradients of electroporation efficiencies that one normal sees. Ideally one would want to know how these data look in terms of electroporated (GFP+) neurons that expressed a specific marker. If for example, if motor neurons are unaffected, one would expect to see a larger percentage of Isl1+/GFP+ than Chx10+/V2+ neurons in these experiments.

Indeed, to inhibit the endogenous activity of E47 and E proteins at large, we opted for a dominant-negative mutant of E47 (E47bm), given that this mutant construct has been reported to act as a dominant-negative over not only E47 but also HEB/TCF12 (as demonstrated by Zhuang et al., 1998, and explained in our manuscript, Subsection “ID2 and E proteins counterbalance each other’s activity during spinal neurogenesis”). We took care of sub-cloning this mutant into a pCIG backbone (which harbors high activity when electroporated *in ovo*) and first confirmed that this E47bm mutant was able to counteract the premature differentiation caused by both E47 and TCF12 (as shown in Figure 3—supplement 1). These latter results led us to consider that E47bm could be used to inhibit the endogenous activity of the different E proteins expressed in the developing spinal cord (and not only E47).

As presented in Figure 4B-F, electroporation of E47bm strongly impaired the generation of neurons derived from progenitors expressing ATOH1 or ASCL1 and to a lesser extent those derived from progenitors expressing NEUROG1, PTF1a or NEUROG2. As explained by the reviewer #1, the results are presented as the ratio of numbers of cells expressing a specific neuronal marker in the electroporated side *vs* the number of cells expressing this marker in the contral-lateral (control) side. In our experience, this type of quantification is classical in studies of spinal neuronal specification (see for instance the work from Jane E. Johnson’s lab, an expert in the field) and presents the advantage of showing results in a very simple and visual manner, when the missexpression generates a strong phenotype as the one caused by E47bm electroporation.

In the revised version of our study, we decided to provide an additional set of data that, we believe, answers the reviewer #1’s concern as well as similar concerns raised later on by the reviewers #2 and #3 (points R2C4, R3C2 and R3C5). In addition to E47bm, we assessed the phenotype caused by *in ovo* electroporation of another dominant-negative construct of E47: E47Δnls-RFP (see the panels G-R from the new Figure 4). This construct of E47 fused to RFP, inserted in a plasmid with low electroporation efficiency, consists of a version of E47 deleted from its nuclear localization signals which thereby impairs the nuclear import of E47, hence its transcriptional activity (Mehmood et al.et al., 2009). As previously reported in vitro, the subcellular localization of this E47Δnls-RFP mutant after *in ovo* electroporation was mostly cytoplasmic (Figure 4G-H’). As seen with E47bm, this E47Δnls-RFP mutant also impaired neuronal differentiation in a cell-autonomous manner (Figure 4G-I). As suggested by the reviewer #1, we analyzed the consequences of E47Δnls-RFP electroporation on the generation of spinal neuron subtypes by quantifying the proportions of differentiated electroporated cells (by focusing on the RFP+ cells that were HuC/D+ or *Sox2*-) that express one of the neuronal subtype markers (Figure 4J-R). Compared to a control plasmid, electroporation of this E47Δnls-RFP mutant reduced about half the proportions of differentiated electroporated cells that express Lhx2/9 or Tlx3 (Figure 4J-M and R), indicating that inhibiting E47 activity hindered the differentiation of spinal progenitors expressing ATOH1 or ASCL1 alone. By contrast, spinal progenitors electroporated with the E47Δnls-RFP mutant efficiently differentiated into Lhx1/5^+^ interneurons or Isl1^+^ motor neurons (Figure 4N-R). Of note, we could even observe Lhx1/5^+^ or Isl1^+^ electroporated cells within the ventricular zone (stars in Figure 4O-O’, Q-Q’), indicative of a premature differentiation of NEUROG1/2/PTF1a-expressing progenitors when E47 activity is impaired. Therefore, the results obtained by inhibiting E47 activity using this new mutant are in agreement with the phenotypic consequences expected by the reviewer #1. They are also in agreement with the results previously obtained with the E47bm mutant, further suggesting that E47 activity is required to avoid NEUROG1/2-expressing progenitors to differentiate prematurely.

R1C9) In Figure 5 the authors provide evidence that the specificity of ASCL1/E47 action are due to its ability to selectively activate sequences containing CAGSTG sequences. E47 also inhibited the ability of NEUROG1 activate its CADATG targets, suggested that E47 can functionally inhibit NEUROG1 at certain sequences.

We understand from this comment that the reviewer #1 was convinced by this part of the study.

R1C10) Finally, the authors tested whether differentiation interactions of ASCL1 and NEUROG1 with E47 might operate in other regions of the CNS. The authors show that E47 enhances the ability of ASCL1 promote neuronal differentiation in the cortex. In contrast they found that E47 blocks the ability of NEUROG1 to promote cortical neuronal differentiation. As with many other experiments in this study, the biological relevance of these interactions are unclear, as we are given no detailed information on the temporal or spatial expression profile of E47 during cortical development.

The idea behind this set of experiments was to try to determine whether the differential cooperation between E47 and ATOH1/ASCL1 and NEUROG1/2 that we had observed in the context of spinal neurogenesis is specific of this region of the developing CNS or if it could be generalized. We believed that the functional assays shown (now in Figure 7I-R and Figure 7—figure supplement 1C) answered this question. It is however true that we did not provide any information about the temporal or spatial expression of E47 in the developing chick cortex. To rectify this shortage of information, we analyzed the expression of *cTcf3/cE2A* by *in situ* hybridization. These new results (presented in Figure 7E-E’ and Figure 7—figure supplement 1B-B”) show that *cTcf3* transcripts are found located in the ventricular zone throughout the D-V axis of the developing chick telencephalon from the beginning of neurogenesis (E3) until at least mid-cortigenesis (E5). This expression pattern is similar to the one described for its murine orthologue during early corticogenesis (Li et al., 2012).

R1C11) The authors have provided detailed information on the statistical methods used in the methods section, and n values for samples in the Figure legends.We understand from this comment that the reviewer #1 is satisfied with the statistical information provided.Reviewer #2:This is a interesting study that puts forward a new mechanistic model to understand how different/limiting levels of E-proteins could have differential effects on neurogenesis through modulating activity of proneural bHLH proteins. The model is supported by manipulating levels of BMP signaling, ID levels, and E-protein/proneural bHLH ratios in the chick neural tube and assessing consequences to division patterns and neuronal subtype specification during embryogenesis. Additional experiments assess the different activities on the proneural bHLH factors with and without the E-protein E47 with respect to transcriptional activity of reporter genes with different motifs. The study is written clearly and data are shown carefully and thoroughly for the most part. The ChIP data are not particularly strong, and the activities rely heavily on over expression assays, nevertheless, multiple types of experiments are used to develop the model and support the conclusions presented.R2C1) The authors come up with the phrase 'dual co-operation of E proteins with proneural proteins' to describe the differential consequences to neurogenesis that result when E-proteins are limiting. It is a confusing description. Based on the model put forward, it is a differential activity of homodimers of ASCL1/ATOH1 versus NEUROG1/NEUROG2 that form when E-proteins are limiting reflecting the intrinsic binding preferences of the different proneural bHLH TFs. To call this differential co-operation of E-proteins does not really help understand what is going on. I recommend changing how this is stated throughout the manuscript.

We totally agree with the reviewer #2 in that the different functional outcomes caused by modulating E proteins’ activity in the ASCL1/ATOH1-expressing versus NEUROG1/2-expressing neural progenitors indeed originate from the differences in intrinsic binding preferences displayed by the distinct proneural proteins per se. We came up with the concept of “dual co-operation of E proteins with proneural proteins” to highlight the notion that E proteins modulate in opposite ways the activities of ASCL1/ATOH1 versus NEUROG1/2. However, we understand the concern of the reviewer #2 as “dual co-operation” might be understood as “dual mechanism of action”, which would indeed be mistaken. Taking this comment into consideration, we thus decided to erase the term “dual co-operation” from our manuscript. To avoid causing ambiguity in the molecular mechanisms at play between the proneural and E proteins, we limited the use of the terms “differential co-operation” throughout the manuscript. In particular, this expression was removed from the subheadings of the Results section and from the title of our study, henceforth entitled “E proteins sharpen neurogenesis by modulating proneural bHLH transcription factors’ activity in an E-box-dependent manner”. We hope that these changes make our message clearer and that they satisfy the reviewer #2’ concerns.

R2C2) Some of the differences in activities could be affected by different levels of the proneural proteins made from the expression constructs. Assessment of the protein levels will be difficult to compare because no tag is included and the antibodies can work at different levels. Minimally it should be considered. Ideally it should be evaluated. This only becomes important because arguments are made about the different effects of E-proteins on the factors. It may be possible that the protein levels of the proneurals are substantially different and are showing differential effects in these assays. This could be true even though pCIG is used for them all.

Indeed, we realized along the course of this study that differences in expression levels of the different proneural and E protein constructs might impact the results obtained and their interpretation. That is why we had decided to sub-clone all these constructs in a similar backbone and thereby get comparable levels of expression of the different proteins (as explained in the Materials and methods section). As shown in the Western blots presented in the Figure 6—figure supplement 3A-C, transfection of the different constructs of ASCL1, NEUROG1 and E47 and their respective hetero- and homodimers (all inserted under the control of the same promoter) yielded the expression of protein amounts that, if not similar, appear at least to be in the same expression range.

To further ensure that differences in the concentrations of electroporation might not alter the functional outcome, we performed dose-dependent analyses for each proneural protein (tested at 0.1 and 1 µg/µl in the developing spinal cord and 0.5 or 1.0 µg/µl in the developing cerebral cortex). These control experiments were presented in the Figure 4—figure supplement 1A and Figure 6—figure supplement 1 (henceforth Figure 5—figure supplement 1 A and Figure 7—figure supplement 1) and explained in the manuscript (subsection “E47 modulates in opposite ways the neurogenic abilities of ASCL1/ATOH1 and NEUROG1/NEUROG2 during spinal neurogenesis” and subsection “E47 modulates in opposite ways the neurogenic abilities of ASCL1 and NEUROG1/NEUROG2 during corticogenesis”). As can be observed, while the concentration did impact on the extent of the effects caused by the different proneural proteins, in no case did it change the way these TFs affected differentiation (always increasing the proportion of electroporated cells that differentiated into neurons). Based on these data, we thus decided to test how the combined electroporation of E47 would affect the effects caused by the sub-optimal concentration of each proneural protein, in order to get a wider range of putative modulation. We apologize for not making it clear enough in the manuscript and have added this precision in the revised version (subsection “ID2 and E proteins counterbalance each other’s activity during spinal neurogenesis”).

In the case of luciferase assays (Figure 6B-E and Figure 6—figure supplement 2B-I), the effects of ASCL1 or NEUROG1 alone were assessed at both 0.5 and 1.0 µg/µl to ensure that the effects caused by E47 addition (electroporated at 0.5 µg/µl) were not simply resulting from an increase in the total concentration.

Therefore, while we agree with the reviewer #2 that differences in the expression levels of the distinct bHLH proteins might somewhat affect the results obtained in terms of strength of effect (promoting more or less differentiation), we believe that we did consider this issue and performed and provided control conditions and experiments which are thorough enough to support the validity of our main findings.

R2C3) Also related to this issue, in the Discussion section it is stated that: "Such co-operation appears to be particularly crucial in the case of ASCL1, whose overexpression could barely increase neurogenic divisions per se." This contrasts with other studies ie Nakada 2014 that show Ascl1 is very efficient at increasing differentiation even at 24 hrs. Is it possible that expression levels are relatively low from the construct used here relative to Nakada, or relative to the other bHLH in the current study as mentioned above? This may need some consideration or ruled out by determining the comparative levels of the overexpressed bHLH proteins.

We could not find a reference “Nakada et al., 2014” in the literature that assessed the function of ASCL1. We believe that the reviewer #2 might be referring to the study by “Nakada et al., 2004” from Jane E. Johnson’s lab in which the authors assessed the consequences of *in ovo* electroporation of ASCL1 (named MASH1 at that time) on neuronal differentiation in the developing chick spinal cord (Nakada et al., 2004, reference already cited in our manuscript). As shown in their Figure 1Q, about 55% of the cells electroporated with ASCL1 were differentiated into neurons after 24 hours (using an ASCL1 construct inserted into a pMiwIII plasmid electroporated at 2 µg/µl as stated in their Materials and methods section). We obtained 45% of EP+;HuC/D+ cells 24 hours after *in ovo* electroporation of ASCL1 at 0.1µg/µl and 57% after 48 hours (Figure 5—figure supplements 1B and C, respectively). We believe that the results obtained in these two independent studies show a comparable efficiency of ASCL1 at increasing differentiation and suggest that the lower proportion of differentiated cells obtained in response to ASCL1 electroporation in our study is a consequence of a diluted concentration of electroporation, and not a defect of the construct we used.

We also bring to the attention of the reviewers another study in which the authors compared the ability of ASCL1 and NEUROG1/2 to induce neuronal differentiation after adenoviral transfection of mouse cortical progenitor cells in vitro(Ge et al., 2006). They reported that infection with NEUROG1/2 caused nearly 100% of cells to differentiate into neurons after 24-48 hours, whereas only ≈25% of the cortical progenitors had differentiated into neurons after infection with ASCL1 (see their Figure 1C). Interestingly, ASCL1 showed an ability to promote migration comparable to that of NEUROG1/2 (see their Figure 1D), arguing against the possibility that the distinct differentiation-inducing capacities of ASCL1 and NEUROG1/2 observed in their assay were due to differences in adenoviral infection efficiencies. This is thus another independent study performed in another neural context that reported, as the authors themselves stated, that ASCL1 has less-potent neurogenic effect than NEUROG1/2 when misexpressed in neural progenitors.

R2C4) All the results in the study are from E47 but the conclusions are generalized to E-proteins. Was HEB tested in these assays? This should be shown if the generalization is being used.

We presented results showing that HEB/TCF12 overexpression promotes neuronal differentiation comparably to E47, and that the effects of both could be rescued by ID2 co-electroporation (Figure 3D-J). As explained in our answer to the point R1C8, we moreover demonstrated that the effects of TCF12 overexpression could be rescued by co-electroporation with the dominant-negative mutant E47bm, which led us to consider that E47bm could be used to inhibit the endogenous activity of the different E proteins expressed in the developing spinal cord.

In this revised version of the manuscript, we moreover provide new results that show that TCF12 addition is able to enhance ASCL1-dependent pKE7:luc activity while inhibiting NEUROG1’s ability to induce pNeuroD:luc activity, similarly to E47 though with milder capacities (Figure 6—figure supplement 2D,E, cited in subsection “E47 modulates the transcriptional activities of ASCL1 and NEUROG1 in an E-box dependent manner and through physical interactions”).

We believe that these results altogether support the notion that E47 and TCF12 modulate the activity of the proneural proteins in a comparable manner. That is why we took the liberty of using the general term “E proteins” in our final model and conclusions. However, if the editor and reviewers consider it more appropriate, we would accept to tone down this generalization in our manuscript and focus our findings on E47.

Reviewer #3:In this study, LeDréau et al., explore how differences in the formation of proneural bHLH-E protein transcription factor complexes can influence neuronal differentiation in the developing spinal cord and cortex. A link to BMP influences on this process is further explored though an examination of the function of Id2, a BMP-regulated inhibitor of proneural bHLH-E protein complexes. Some of the initial observations include findings that shRNA knockdown of Bmp7, Smad1, or Smad5 potently suppresses the formation of certain classes of spinal neurons (dI1, dI3, and V2a/b) while having less of an effect on other classes (dI6, V0, V1, and MNs). These trends are argued to align with the association of these two groups with the activity of the proneural factors Ascl1 and Atoh1 vs. Neurog1 and Neurog2. The connection of BMP signaling to proneural genes is supposed by evidence that BMP7/Smad1/5 can regulate Id2. shRNA knockdown of Id2 appears to broadly enhance neurogenic differentiation.The focus of the study then shifts to an exploration of the requirement and sufficiency of E proteins, primarily E47, for differentiation of different classes of neurons. The key finding is an unexpected discrepancy in how E47 modulates the neurogenic activity of Ascl1/Atoh1 vs. Neurog1/Neurog2. E47 potentiates the activities of Ascl1/Atoh1 while showing an inhibitory effect on Neurog1/Neurog2 in some assays. It is further argued that Neurog1/Neurog2 but not Ascl1/Atoh1 may primarily function as homodimeric complexes and that E47 addition may shift the capacity of the bHLH proteins to bind to certain E box motifs, favoring interactions with CAGSTG sites vs. CADATG sites. Lastly, the authors show that the differential cooperativity of Ascl1 and E47 vs. Neurog1/2 and E47 also holds true in the embryonic cortex, suggesting that this may be a widespread phenomenon.Overall, the study makes some interesting and important observations. First, E proteins do not universally increase the activity of proneural bHLH proteins, a presumption that comes from classic studies of the cooperativity of myogenic bHLH factor and E proteins. Second, the study argues that certain proneural bHLH proteins such as Neurog1 and Neurog2 may function as homodimeric complexes while others are heterodimeric, which has implications for whether the complex could be regulated by Id proteins.R3C1) However, the study also has significant weaknesses, most notably inconsistencies with some results and the conclusions/models that are drawn. In addition, while it is suggested that the DNA binding specificity of Neurog1/2 may be altered by the addition of E47, conclusive proof is not provided.

To demonstrate that E47 restrains the DNA binding and transcriptional activity of NEUROG1 on CADATG motifs, we presented results obtained from luciferase (Figure 6E, Figure 6—figure supplement 2C) and ChIP assays (Figure 6G). In the revised version presented herein, we bring novel results obtained using TCF12 (Figure 6—figure supplement 2E), an additional luciferase reporter (Figure 6—figure supplement 2G) and new experimental conditions presenting the effects of ID2 addition on top of NEUROG1 and E47 (Figure 6—figure supplement 2I). To further demonstrate that these results were due to a direct/physical interaction between E47 and NEUROG1, we provided luciferase assays in which we compared the activity of tethered constructs of NEUROG1 homodimers and NEUROG1-E47 heterodimers (Figure—figure supplement 3E). We moreover provided results showing the phenotypical consequences of this modulation of NEUROG1/2 transcriptional activity by E47 on the mode of division (using the pSox2:luc, pSox2GFP and pTis21:RFP reporters as shown in the Figures henceforth renamed 5L-P and Figure 5—supplement 1D), the final EP+ cell number (Figure 5I) and the proportion of neuronal differentiation (Figure 5E,F and Figure 5—supplement 1B, C) as well as results obtained with the tethered NEUROG1-NEUROG1 and E47-NEUROG1 dimers (Figure 6K-N). We sincerely believe that these pieces of evidence put altogether are consistent and strongly support our conclusions.

R3C2) Lastly, the studies showing the importance of E47 for neurogenesis are based on misexpression studies without complementary loss of function data. It thus remains unclear whether Neurog1/Neurog2 functions are indeed independent of E protein activities as proposed.

As explained in details in the answer to the point R1C8, we had opted for a dominant-negative strategy to inhibit the endogenous activity of E47 and E proteins at large, based on the literature suggesting that the consequences of E47 loss-of-function might have been masked and compensated for by the activity of other E proteins expressed in our system (Zhuang et al., 1998). In our opinion, the results obtained with the E47bm mutant (shown in Figure 4B-F) are in agreement with the phenotype expected from our hypothesis.

In addition, we now provide a new set of experiments performed with another dominant-negative E47 mutant (E47Δnls-RFP), as explained in details in our answer to the point R1C8 (please head for this section to read the description of these new results). Although these results represent another misexpression strategy instead of a loss of function strategy as asked by the reviewer #3, we believe that the results obtained and presented in the Figure 4G-R reinforce our conclusions, and we hope that they will satisfy the concerns of the reviewers #1 and #3.

R3C3) A central premise of the study is that BMP signaling differentially regulates neurogenesis by controlling Id2 expression. However, the Id2 shRNA experiments in Figure 2 seems to show broader effects of Id2 loss encompassing both Ascl1/Atoh1 and Neurog1/Neurog2-associated groups. How does one reconcile this discrepancy with the overall model?

We agree with the reviewer #3 on the fact that sh-Id2 *in ovo* electroporation affected one particular neuronal subtype deriving from NEUROG1-expressing progenitors more than sh-Bmp7 or sh-Smad1/5 (V0v neurons specifically, Figure 2O) and less markedly several neuronal subtypes deriving from ASCL1/ATOH1-expressing progenitors (Figure 2O). Nevertheless, the overall tendencies of these phenotypes are similar enough to be noticed (Figure 2O). It actually surprised us that the phenotype caused by sh-Id2 would reproduce the overall phenotype caused by BMP7 or SMAD1/5 inhibition. We were expecting that the reduced activity of ID2 might have been compensated for by the activity of other IDs, which are expressed in overlapping D-V progenitor domains (see Annex 6).

As explained in our answer to the point R1C5, results from our previous study indeed demonstrated that *cId1, cId2* and *cId3* are regulated by the canonical BMP pathway during spinal neurogenesis (see Annex 6) and brought evidence that at least *cId2* and *cId3* (it was less evident for *cId1*) might represent direct targets of SMAD1/5 mediating their function on the control of the mode of division of spinal progenitors (see panels H-I from Annex 3). In order to get a proof of concept that the ID factors mediate the function of the canonical BMP pathway during spinal neurogenesis, we had chosen to focus on ID2 for the reasons mentioned above and the fact that *cId2* presents an expression pattern broader than *cId3* for instance. Our intention was to say that ID2 represents one of the factors regulated by the canonical BMP activity and involved in mediated BMP function during spinal neurogenesis, not that ID2 is the sole factor involved. We apologize if we did not make this point clear enough. So, differences in the phenotypes obtained with sh-Id2 as compared to the ones obtained with sh-Bmp7/Smad1/5 are likely to be due to the fact that the canonical BMP activity is mediated not only by ID2 but by additional targets of the pathway (possibly other IDs).

On the other hand, it is also very likely that the expression of *cId2* is not regulated solely by the canonical BMP pathway and is probably modulated by additional signaling pathways during spinal neurogenesis. For all these reasons, we did not expect that the phenotype caused by ID2 loss of function would match 100% the one caused by BMP7 or SMAD1/5. However, we believe that they are similar enough to support the idea that ID2 acts downstream of the canonical BMP pathway.

R3C4) The inhibitory effects of E47 expression appear to be only transient- in Figures 4K and 4M, suppression is seen at the 24 h time point, but not at 48 h. How does this difference play into the central model?

Rather than the effect of E47 being only transient, the fact that the inhibitory effects of E47 on NEUROG1-induced differentiation are observed after 24 hours and not anymore after 48 hours results from the way NEUROG1, alone or in combination with E47, affects the mode of division of spinal progenitors (Figure 5L-P). To make our explanation more explicit, we propose you a simulation of the effects of NEUROG1 overexpression alone or combination with E47 on the progressive differentiation of the pool of electroporated cells (Annex 8).

Let assume that 10 spinal progenitors (white circles) have integrated the plasmid at the time of *in ovo* electroporation. Under the effect of NEUROG1 overexpression, these 10 progenitors will undergo symmetric proliferative (PP, green circle), asymmetric (PN, yellow circle) or self-consuming neurogenic divisions (NN, red circles) according to the proportions assessed with the pSox2:GFP and pTis21:RFP reporters (results presented in Figure 5P). The outcome of this first round of division results in the generation of 7 new progenitors (resulting from the PP and PN divisions) and 13 daughter cells committed to neuronal differentiation (resulting from the PN and NN divisions, black circles). By definition, these differentiating neurons won’t divide anymore. As specified in the bottom of the panel A, this first round of division generated 20 cells in total, of which 13 are already differentiating. This represents a proportion of EP+ cells that differentiated into neurons of 65%, a proportion that is pretty close to the one quantified 24 hours after NEUROG1 overexpression (Figure 5E). We are thus left with 7 electroporated progenitors for the second round of division, which should all undergo PN or NN divisions based on the proportions of PP/PN/NN applied previously. By following on with this simulation, after a fourth round of divisions we are left with only 1 progenitor cell for a total of 33 electroporated cells present in the system. This represents a proportion of EP+ cells that differentiated into neurons of 97%, a proportion that matches also pretty well the one quantified 48 hours after NEUROG1 overexpression (Figure 5E).

If we apply the same modeling for the co-electroporation of NEUROG1 and E47, the proportions of PP/PN/NN divisions are slightly shifted towards proliferative divisions at the beginning. After the first round of division, we would obtain 9 progenitors and 11 differentiating neurons, representing a proportion of EP+ cells that differentiated into neurons of 45%. Again, this proportion obtained by simulation is pretty close to the one quantified (Figure 5E). At the end of the fourth round, the proportion of differentiated cells among the electroporated cells reaches 93%, which again is pretty similar to the one quantified (Figure 5E). It is also interesting to note that although we started with the same number of spinal progenitors electroporated at the beginning and applied the same experimental duration, the final numbers of EP cells are markedly different, in agreement with the experimental results described (Figure 5I). When E47 is combined with NEUROG1, more progenitor daughter cells have been generated and divided throughout the rounds of division, which fits also very well with the results obtained with the Violet assay (Figure 5K). Altogether, these results illustrate how a slight shift in the proportions of the 3 different modes of divisions can impact the final numbers of neurons that are generated and the pace of differentiation.

We hope that these simulations will help convincing the reviewers of the logic and consistency of the data we presented in Figure 4 (now Figure 5).

R3C5) Evidence that E47 inhibits Neurog1/2 function solely comes from gain of function experiments, which can be artifactual. The authors should examine the consequences of E47 loss using shRNA or CRISPR mutation approaches. If it is true that Neurog1/2 act as homodimers, then there should be little change in the differentiation of the Neurog1/2-associated neuron groups, while Ascl1/Atoh1-associated groups should be severely impacted.

We previously detailed in our answers to the points R1C8 and R3C2 the reasons that led us to opt for a dominant-negative strategy over the use of a sh-RNA targeting *cE47*. We wish to highlight the fact that the consequences that the reviewer #3 would expect from the use of a sh-RNA are fitting exactly with the phenotype obtained with the E47bm mutant (shown in Figure 4B-F). The new results obtained with the E47Δnls-RFP mutant fit also well with these predictions, with the additional demonstration that a fraction of the electroporated spinal progenitors from NEUROG1/2/PTF1a-expressing domains differentiated prematurely (Figure 4O, Q). These latter results support the notion that within these domains E47 acts to restrain NEUROG1/2 activity in order to prevent a premature differentiation.

As discussed in our manuscript (Discussion section), a recent study reported the consequences of the genetic depletion of E47 on mouse cortical development (Pfurr et al., 2017). This study revealed that E47 depletion resulted in an increased production of cortical excitatory neurons at mid-corticogenesis. Considering that it is assumed that NEUROG2 constitutes the main proneural protein at play during early corticogenesis, these results are consistent with our results and model. They moreover suggest that inhibiting E47 function, whether this is achieved by a loss of function strategy or missexpresion of a dominant-negative mutant, can produce comparable phenotypes.

R3C6) The stoichiometry of various proneural proteins and E protein needs to be carefully considered- too much of either the proneural factor or E protein could throw off activities. Ideally, the amount used should be titrated to identify the optimal amounts to be used. It might be further useful to judge whether equivalent amounts of each protein are present in the experiments using epitope tagged versions of the proteins and immunoblotting (or semi-quantitative fluorescence microscopy).

The reviewer #2 emitted a similar concern (points R2C2, R2C3). Please head for these paragraphs to read our answer.

R3C7) The authors show that another E protein, Tcf12 is also expressed in the spinal cord, yet few experiments are performed to explore its relevance. Does Tcf12 similarly provide enhancement of Ascl1/Atoh1 function while inhibiting Neurog1/2?

In this revised version of the study, we provide additional data showing that TCF12 addition is able to enhance ASCL1-dependent pKE7:luc activity while inhibiting NEUROG1’s ability to induce pNeuroD:luc activity, similarly to E47 though with milder capacities (Figure 6—figure supplement 2D,E, cited in subsection “E47 modulates the transcriptional activities of ASCL1 and NEUROG1 in an E-box dependent manner and through physical interactions”). We believe that these results, together with the results previously presented, support the notion that E47 and TCF12 modulate the activity of the proneural proteins in a comparable manner. However as previously said, if the editor and reviewers consider it more appropriate, we would accept to tone down this generalization in our manuscript and focus our findings on E47.

R3C8) The data collectively suggest that the presence of E47 alters the preferences of proneural bHLH proteins for certain E box motifs, yet direct proof remains lacking. These conclusions could be strengthened by the inclusion of DNA binding experiments (EMSA) showing that the mixing of proneural proteins with E47 changes their affinity for certain E boxes as is implicated in the Figure 7 model. Id proteins could also be added in to test whether certain complexes (such as Neurog1/2 homodimers) are indeed resistant to Ids. Another way to achieve these results might be to build out the use of reporter constructs adding modified versions of the pkE7 construct containing various E box motifs and ask whether there are switches in the activity levels of the different reporters.

As already answered in the point R3C2, we sincerely believe that we brought an amount of data convincing enough to support the notion that E47 modulates the transcriptional activity of ATOH1/ASCL1 and NEUROG1/2 in an E-box-dependent manner. However, we agree with the reviewer #3 that his/her suggestions of additional experiments would reinforce the conclusions of our work and the model presented. For this reason, in the revised version of our study we provide additional data in line with the reviewer #3’s suggestions, and which are included in the revised version of the Figure 6—figure supplement 2.

To strengthen the notion that the E-box context determines the way E proteins modulate the activity of the distinct proneural proteins, luciferase assays were performed using two additional reporters: pDll1-M and pDll1-N, which have been previously described to respectively respond to ASCL1 and NEUROG2 (Castro et al.et al., 2006) and are based on conserved regulatory elements found in the promoter of the gene *Δ-like1* (Beckers et al.et al., 2000). The promoters of these pDll1-M:luc and pDll1-N:luc reporters contain 3 CAGSTG + 1 CADATG and 1 CAGSTG + 3 CADATG motifs, respectively (Figure 6—figure supplement 2A). In the context of pDll1-M:luc activity, the addition of E47 had only an additive effect to ASCL1 activity (Figure 6—figure supplement 2F), compared to the synergistic effect observed on pKE7:luc activity (Figure 6B). In the context of pDll1-N:luc activity, addition of E47 also inhibited the induction caused by NEUROG1 (Figure 6—figure supplement 2G), though with milder effects compared to the strong repression observed on pNeuroD:luc activity (Figure 6E). These results reinforce our model whereby the outcome of the co-operation of E47 with the distinct proneural TFs depends on the balance between the CAGSTG and CADATG motifs present in each DNA target region and on the intrinsic binding preferences of the proneural proteins per se.

In addition, we present luciferase assays that include experimental conditions in which ID2 was combined with E47 and ASCL1 or NEUROG1. Addition of ID2 partially rescued both the synergistic effect of E47 on pKE7:luc activity when combined with ASCL1 and the inhibitory effect of E47 on NEUROG1-induced pNeuroD:luc activity (Figure 6—figure supplement 2H,I). These results are in agreement with the notion that IDs act by sequestering E proteins from their interaction with proneural proteins, and reinforced the notion that the context-dependent modulatory effect of E47 on ASCL1 or NEUROG1 activities relies on physical interactions.

We hope that these additional results will satisfy the reviewer #3’s curiosity.

R3C9) The ChIP studies shown in Figure 5 are performed in 293 cells, while all of the luciferase assays and cell differentiation experiments are conducted in the chick spinal cord or brain. It would be best to perform ChIP in a system where some signs that the proneural bHLH factors with or within E47 are transcriptionally active.

We wish to emphasize, as the reviewer #3 recognized himself/herself, that all the results presented in our study, except for the ChIP data, were obtained from in vivo experiments.

As an in vitro system, in contrast to the opinion of the reviewer #3, we actually preferred to use a cell line in which no endogenous proneural protein would be expressed to avoid the possibility that these endogenous proneural proteins might interfere with the designed experiments and bias the results’ interpretation. We believe that the results obtained in these ChIP assays are supporting the results obtained in vivo.

R3C10) The studies here do not address the secondary wave of proneural proteins, i.e. Neurod1 and Neurod4 which come on as cells begin to differentiate. Is the expression and/or function of these factors also impacted by E47 presence/absence and Id2 inhibition?

We agree with the reviewer #3 that, in view of our results, studying how the E proteins and ID factors modulate the activity of NEUROD1/4 would be very interesting. As explained by the reviewer #3, NEUROD1/4 represent another group of bHLH proteins that act downstream of the proneural proteins to promote neuronal differentiation (Guillemot, 2007). But for this very same reason, we truly believe that this question is beyond the scope of our study and should rather be tackled in future studies.

Additional References:

Ge, W., He, F., Kim, K.J., Blanchi, B., Coskun, V., Nguyen, L., Wu, X., Zhao, J., Heng, J.I., Martinowich, K. et al. 2006. Coupling of cell migration with neurogenesis by proneural bHLH factors. Proc Natl Acad Sci U S A 103(5): 1319-1324.

Tozer, S., Le Dreau, G., Marti, E., and Briscoe, J. 2013. Temporal control of BMP signalling determines neuronal subtype identity in the dorsal neural tube. Development 140(7): 1467-1474.